# ErbB expressing Schwann cells control lateral line progenitor cells via non-cell-autonomous regulation of Wnt/β-catenin

Mark E Lush[1,2], Tatjana Piotrowski[1,2]*

[1]Stowers Institute for Medical Research, Kansas City, United States; [2]Department of Neurobiology and Anatomy, University of Utah School of Medicine, Salt Lake City, United States

**Abstract** Proper orchestration of quiescence and activation of progenitor cells is crucial during embryonic development and adult homeostasis. We took advantage of the zebrafish sensory lateral line to define niche-progenitor interactions to understand how integration of diverse signaling pathways spatially and temporally regulates the coordination of these processes. Our previous studies demonstrated that Schwann cells play a crucial role in negatively regulating lateral line progenitor proliferation. Here we demonstrate that ErbB/Neuregulin signaling is not only required for Schwann cell migration but that it plays a continued role in postmigratory Schwann cells. ErbB expressing Schwann cells inhibit lateral line progenitor proliferation and differentiation through non-cell-autonomous inhibition of Wnt/β-catenin signaling. Subsequent activation of Fgf signaling controls sensory organ differentiation, but not progenitor proliferation. In addition to the lateral line, these findings have important implications for understanding how niche-progenitor cells segregate interactions during development, and how they may go wrong in disease states.

*For correspondence: pio@stowers.org

**Competing interests:** The authors declare that no competing interests exist.

**Reviewing editor**: Tanya T Whitfield, University of Sheffield, United Kingdom

## Introduction

The cell signaling events that govern progenitor quiescence, activation and differentiation are incompletely understood, but emerging data in many tissues indicate that dynamic interactions between progenitors and specialized niche environments play key roles in regulating the properties of progenitor pools. Hence, understanding niche-progenitor interactions at the cellular level is crucial for building a general understanding of this process. The development of the zebrafish lateral line is an excellent model system to study progenitor cell regulation, as it consists of relatively few cells that are easily accessible and amenable to experimental manipulations.

The sensory organs of the lateral line are called neuromasts. Neuromasts contain support and mechanosensory hair cells that detect water motion. The first set of neuromasts is laid down by a migrating primordium (primI) that develops from a placode just posterior to the otic vesicle. As the primordium migrates posteriorly along the trunk of the embryo it deposits five to six primary neuromasts and a chain of interneuromast cells that connects each neuromast (*Ghysen and Dambly-Chaudiere, 2007*). Before the placode becomes migratory, its anterior portion splits off and forms the posterior lateral line ganglion (*Northcutt and Brandle, 1995*). Lateral line axons closely follow the migrating primordium and eventually innervate deposited neuromasts (*Gilmour et al., 2004*). In turn, neural crest-derived Schwann cells migrate along the axons which they eventually myelinate (*Gilmour et al., 2002*; *Lyons et al., 2005*). Thus, interneuromast cells, axons and Schwann cells are in close contact during the early stages of lateral line development (see diagram in *Figure 1A*; *Whitfield, 2005*).

**eLife digest** All the different types of cells that make up the body of an animal are descended from a single fertilized egg. As this egg develops into an embryo, the cells divide and specialize to become a specific type of cell, such as: a liver cell, a muscle cell or a nerve cell. The cells in the embryo that are destined to become specific cell types are called progenitor cells. However, these cells are also found within adult tissues, where they wait until they are needed to replace old or damaged cells.

Zebrafish are commonly used in scientific research and, like other fish, they have a 'lateral line' that runs along both sides of the body and contains cells that detect movements in the surrounding water. During its development, the lateral line contains many progenitors that are primed to form more of these sense organs. The lateral line is also connected to nerve cells that relay information about water movements to the central nervous system, while other cells called Schwann cells support the nerve cells. The local environment or 'niche' created by the Schwann cells is known to prevent the progenitor cells within the lateral line from becoming their specific cell type too early. However, the molecules that cause progenitor cells to stop dividing, and later restart dividing and change in to their predestined cell type is not well understood.

Now Lush and Piotrowski have discovered that signaling through a protein called ErbB causes the Schwann cells to multiply, but has the opposite effect on nearby progenitor cells in the lateral line. ErbB signaling in the Schwann cells inhibited various signaling pathways in the progenitor cells; and whilst some of these pathways normally encourage the progenitors to multiply, others cause them to change into their specific cell type.

The findings of Lush and Piotrowski have important implications for understanding how the interactions between progenitor cells and the cells around them affect their development. These findings may be useful for understanding diseases caused when the control of cell multiplication or cell-type changes goes awry—such as developmental abnormalities or cancer.

The adult posterior lateral line contains many more neuromasts than the 7–8 neuromasts initially laid down by primI. These additional 'secondary' neuromasts originate from several sources. (A) A second primordium, primII, develops at 40 hr post fertilization (hpf) and deposits neuromasts in between the previously deposited sensory organs (*Sapede et al., 2002*; *Nunez et al., 2009*). (B) Intercalary neuromasts arise during the first 2 weeks of development by proliferation and differentiation of primI deposited interneuromast cells (*Sapede et al., 2002*; *Grant et al., 2005*; *Lopez-Schier and Hudspeth, 2005*; *Nunez et al., 2009*). and (C) During juvenile stages neuromast stitches arise through budding from primary neuromasts (*Ledent, 2002*; *Wada et al., 2013a*).

We and others have previously shown that Schwann cells play a crucial role in negatively regulating the timing of differentiation of interneuromast cells into intercalary neuromasts (*Grant et al., 2005*; *Lopez-Schier and Hudspeth, 2005*). In zebrafish that lack Schwann cells along the lateral line, such as in mutants for *sox10* and the ErbB pathway members *erbb2*, *erbb3b* and *nrg1-3,* intercalary neuromasts form precociously (*Grant et al., 2005*; *Rojas-Munoz et al., 2009*; *Perlin et al., 2011*). As Schwann cells require axons for migration along the lateral line, *neurogenin* mutants that lack a posterior lateral line ganglion, also show extra neuromasts (*Lopez-Schier and Hudspeth, 2005*). Likewise, extra neuromasts form after posterior lateral line ganglion extirpation or Schwann cell ablation (*Grant et al., 2005*; *Lopez-Schier and Hudspeth, 2005*). These experiments suggest that Schwann cells contribute to an inhibitory niche that keeps lateral line progenitor cells from undergoing precocious proliferation and differentiation.

The signaling pathways that orchestrate intercalary neuromast formation are currently unknown. In contrast, the early development of the migrating lateral line has been extensively studied. Complex cell signaling interactions between Wnt/β-catenin, Fgf, Notch and chemokine pathways regulate proliferation, neuromast formation and migration (*Aman and Piotrowski, 2009*; *Ma and Raible, 2009*; *Chitnis et al., 2012*). Wnt/β-catenin signaling in the leading region of the primordium initiates and restricts Fgf signaling to the trailing region. In turn, Fgf signaling upregulates *dkk1b*, a secreted Wnt/β-catenin inhibitor, that restricts Wnt/β-catenin signaling to the leading region (*Aman and Piotrowski, 2008*). Fgf signaling induces apical constriction in clusters of cells resulting in

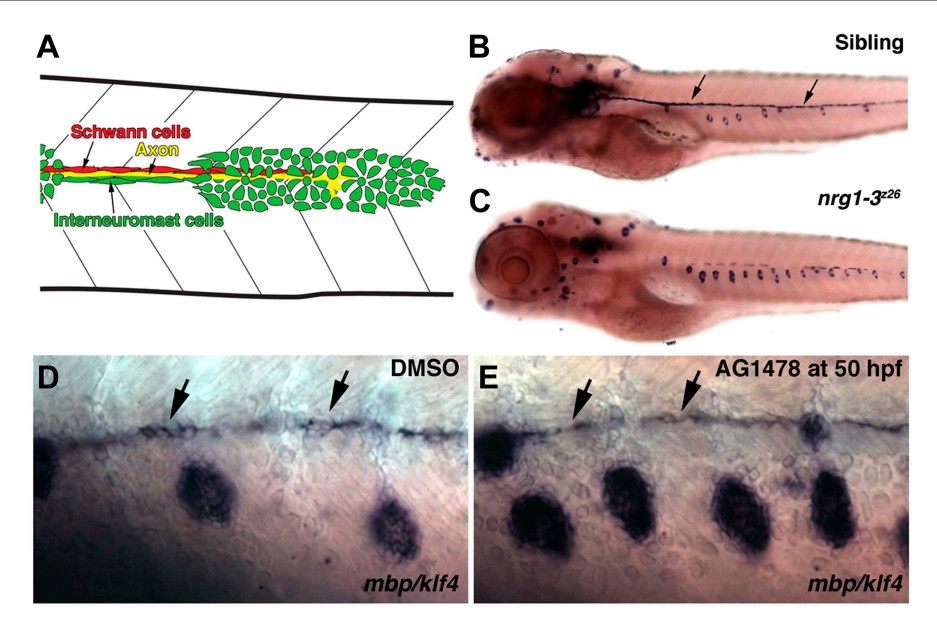

**Figure 1**. Illustration of cell types in the migrating lateral line. (**A**) As the primordium migrates it deposits neuromasts and a chain of interneuromast cells (green cells). Pioneer axons (yellow line) of the posterior lateral line ganglion grow out with the primordium. Schwann cells (red cells) migrate and proliferate along axons. *nrg1-3^{z26}* mutants and pharmacological inhibition of ErbB signaling mimics the *erbb* phenotype. (**B**–**E**) Double in situ hybridization was performed to label Schwann cells with *myelin basic protein* (*mbp*) and neuromasts with *klf4* at 5 dpf. (**B**) Control siblings with Schwann cells (arrows) along the lateral line nerve and normal neuromast number. *nrg1-3^{z26}* mutants mimic *erbb2* and *erbb3b* mutants in that they lack Schwann cells along the lateral line and have increased neuromast number (**C**). The brown cells along the midline in both sibling and *nrg1-3^{z26}* are pigment cells. (**D** and **E**) Double in situ hybridization for *mbp* and *klf4* in DMSO or AG1478 treated larvae from 50 hpf. Compared to DMSO treatment (**D**), increased neuromasts are seen in AG1478 treated larvae (**E**). *mbp* expression along the midline shows that Schwann cells (arrows) are still present at 5 dpf when AG1478 was given at 50 hpf (**E**), compare to DMSO treated (**D**).

The following figure supplements are available for figure 1:

**Figure supplement 1**. Mutations in the *erbb* signaling pathway show precocious neuromast formation by 5 dpf.

**Figure supplement 2**. *nrg1-3^{z26}* mutants have defects in adult pigment pattern.

**Figure supplement 3**. *nrg1-3^{z26}* mutants lose neuromasts as they age.

**Figure supplement 4**. ErbB inhibition after lateral line migration is complete causes a decrease in proliferation and number of lateral line Schwann cells.

the morphogenesis of rosette shaped protoneuromasts (*Lecaudey et al., 2008*; *Nechiporuk and Raible, 2008*). Fgf signaling is also required for hair cell differentiation (*Millimaki et al., 2007*; *Nechiporuk and Raible, 2008*), and both Wnt/β-catenin and Fgf signaling are required for proliferation within the migrating primordium (*Aman et al., 2011*).

This study focuses on the development of intercalary neuromasts to elucidate the molecules that regulate progenitor cell proliferation and development. We characterize the signaling pathways required for precocious intercalary neuromast formation downstream of ErbB signaling. In the absence of Schwann cells, or ErbB/Neuregulin signaling, Wnt/β-catenin and Fgf signaling are increased. Wnt/β-catenin signaling is required for interneuromast proliferation while Fgf signaling is required for subsequent rosette formation and cellular differentiation. Schwann cells maintain interneuromast cells as quiescent progenitors by expressing a, as yet unidentified Wnt/β-catenin inhibitor. These

findings illustrate the intricate manner in which diverse signaling pathways coordinate distinct aspects of the niche-progenitor interaction needed to maintain the proper balance and timing of this dynamic cell population.

## Results

### Mutations in ErbB/Neuregulin pathway members cause precocious differentiation of intercalary neuromasts

Intercalary neuromasts arise during a 2-week period from interneuromast cells, which are initially deposited by primI as a chain of single cells in between primary neuromasts (*Grant et al., 2005*). The cellular relationships within the migrating lateral line are outlined in *Figure 1A*. Deposited interneuromast cells are initially in close contact with Schwann cells (*Figure 1A*, green and red cells respectively). A variety of lines of genetic evidence, including our own, demonstrates that ErbB signaling plays a fundamental role in the migration of Schwann cells that control the proliferation of interneuromast cells. Mutations in ErbB receptors (*row/erbb2*, *hps/erbb3b*) cause a loss of Schwann cells along the lateral line nerve leading to precocious interneuromast proliferation and intercalary neuromast differentiation (*Figure 1—figure supplement 1A–E*; *Grant et al., 2005*; *Lyons et al., 2005*; *Rojas-Munoz et al., 2009*). We recently identified a mutation in the ErbB ligand *neuregulin 1-3 (nrg1-3$^{z26}$)* that also lacks Schwann cell migration along lateral line axons (*Perlin et al., 2011*), and forms supernumerary neuromasts (*Figure 1B–C*). *nrg1-3$^{z26}$* mutants survive to adulthood but exhibit an adult pigment pattern and neuromast degeneration phenotype (*Figure 1—figure supplement 2,3*), similarly to *erbb3b* adult mutant fish (*Budi et al., 2008*; *Honjo et al., 2011*). Below we identified in which cell types different members of the ErbB/Neuregulin pathway are functioning to control Schwann cell migration and lateral line progenitor proliferation and differentiation.

### Pharmacological inhibition of ErbB signaling mimics the *erbb2/3b* mutant phenotype

During development, signaling pathways are repeatedly employed. We therefore wanted to test if the extra neuromast phenotype is due solely to loss of Schwann cells along the lateral line, or if ErbB signaling plays an additional role in inhibiting proliferation of interneuromast cells. Therefore, ErbB signaling was inhibited with the ErbB tyrosine kinase inhibitor AG1478 (*Osherov and Levitzki, 1994*), before (24 hpf) and after (48 hpf) completion of Schwann cell migration, and neuromast number was assessed at 5 days post fertilization (dpf). As expected, inhibition of ErbB signaling at 24 hpf, when Schwann cells migrate, leads to a loss of Schwann cells and the formation of extra neuromasts (*Figure 1—figure supplement 1F*; *Rojas-Munoz et al., 2009*). Interestingly, ErbB inhibition is able to increase neuromast numbers even in the presence of Schwann cells, if supplied between 50–59 hpf (*Figure 1D–E*, *Figure 1—figure supplement 1F*). The presence of Schwann cells is based on detection of *myelin basic protein* (*mbp*) expression (*Figure 1D–E*, arrows). These data suggest that ErbB signaling not only regulates Schwann cell migration but also plays a continued role in post-migratory Schwann cells in inhibiting interneuromast cell proliferation.

A potential caveat for that interpretation is that ErbB signaling is also required for Schwann cell proliferation (*Lyons et al., 2005*; *Raphael et al., 2011*), and pharmacologically lowering the number of Schwann cells could secondarily affect interneuromast proliferation. To test when Schwann cell numbers are reduced upon ErbB inhibition at 48 hpf we used the *Tg(foxd3:gfp)* zebrafish line that expresses EGFP in neural crest derived tissues including Schwann cells (*Gilmour et al., 2002*). Using BrdU labeling in control and AG1478 treated *Tg(foxd3:gfp)* fish, we counted BrdU positive *Tg(foxd3:gfp)* Schwann cells at 6, 14 or 24 hr post treatment. ErbB inhibition induces a decrease in BrdU incorporation in Schwann cells at 6 hr post treatment, however the total Schwann cell number remains unchanged (*Figure 1—figure supplement 4A–B*). A reduction in Schwann cell proliferation continues at 14 and 24 hr post treatment, at which point it is accompanied by a decrease in total Schwann cell numbers (*Figure 1—figure supplement 4C–F*). The finding that Schwann cell numbers are not affected at 6 hr post treatment is important, as the first molecular changes in interneuromast cells are already observed at this stage (see below, 'Wntβ-catenin signaling activation occurs prior to Notch and Fgf activation within interneuromast cells after ErbB inhibition'). This suggests that ErbB signaling affects lateral line proliferation directly, rather than indirectly via the regulation of Schwann cell number. Thus, ErbB signaling has independent functions in Schwann cell migration and lateral line progenitor proliferation and

differentiation. To elucidate if ErbB signaling controls progenitor proliferation cell-autonomously or non-cell-autonomously, we performed transplantation experiments between mutant and wild type embryos.

## ErbB2 is required in Schwann cells and Nrg1-3 is required in lateral line axons to inhibit precocious formation of intercalary neuromasts

Transplantation experiments revealed that *erbb3b* and *sox10* act cell-autonomously in Schwann cells to regulate their migration and inhibit precocious interneuromast proliferation (*Grant et al., 2005*). As *erbb2* is ubiquitously expressed and AG1478 blocks ErbB signaling globally, we wanted to clarify in which cell type ErbB2 signaling is required. We transplanted dextran-Alexa568 labeled cells from *erbb2* mutant blastomere stage donor embryos into wild type *Tg(foxd3:gfp)* host embryos and analyzed clones that gave rise to interneuromast cells. *erbb2* mutant interneuromast cell clones failed to produce extra neuromasts by 4 dpf, suggesting that ErbB2 signaling is not required in interneuromast cells to prevent intercalary neuromast formation (*Figure 2A–A'*, n = 0/4). On the other hand, when transplanted *Tg(foxd3:gfp)* wild type cells gave rise to Schwann cell clones in an *erbb2* mutant host embryo, the extra neuromasts phenotype at 4 dpf was rescued (*Figure 2B–B'''*, n = 9/9). Rescue was only achieved when Schwann cell clones extended all the way along the trunk by 48 hpf. Transplanted cells that only gave rise to interneuromast cells, or a few Schwann cells that did not reach the tail tip, failed to rescue the *erbb2* mutant phenotype (n = 0/12). These transplant experiments illustrate that, similar to ErbB3b, ErbB2 is required in Schwann cells to inhibit intercalary neuromast formation.

We confirmed these results genetically by generating a transgenic zebrafish line that drives human dominant-negative ErbB4 (DNErbB4) in neural crest-derived tissues, including Schwann cells, using the *sox10* promoter. DNErbB4 blocks Neuregulin-induced signaling in cell culture and in vivo (*Rio et al., 1997*; *Chen et al., 2006*). We isolated a stable transgenic line for analysis designated *Tg(sox10:DNhsaerbb4-rfp)*, from now on called *Tg(sox10:DNerbb4)*. As expected, in transgenic embryos Schwann cells fail to migrate along the lateral line and they develop extra neuromasts (data not shown and *Figure 2C–D*). Transgenic fish survive to adulthood, are fertile and exhibit no other obvious phenotypes. Along with the transplantation experiments these results demonstrate that ErbB2/3b signaling is required in Schwann cells in order to non-cell-autonomously regulate neuromast number.

*nrg1-3* is expressed in lateral line ganglia suggesting that it is required in lateral line axons (*Perlin et al., 2011*). To functionally test in which cell type Nrg1-3 is required we performed transplantation experiments between wild type and *nrg1-3$^{z26}$* embryos. As donors we used *Tg(cldnb:lyngfp)* embryos that express EGFP in all lateral line cells including the ganglion (*Haas and Gilmour, 2006*). We rescued the extra neuromast phenotype in *nrg1-3$^{z26}$* mutants with transplanted wild type cells that gave rise to large posterior lateral line ganglion clones (*Figure 2E–E'''*, n = 13/13). Transplanted cells that only contributed to interneuromast cells or to few lateral line ganglion neurons failed to rescue *nrg1-3$^{z26}$* (n = 0/19). This is consistent with prior findings that wild type posterior lateral line ganglion clones rescue Schwann cell migration in *nrg1-3$^{z26}$* mutant embryos (*Perlin et al., 2011*). Thus, the ligand Nrg1-3 is required in axons to induce migration and proliferation of ErbB expressing Schwann cells and inhibit precocious formation of intercalary neuromasts.

Combined, these experiments revealed that the quiescent niche consists of axonal, membrane bound, Nrg1-3 that signals to ErbB receptors within Schwann cells. In response, Schwann cells send a signal to interneuromast cells that inhibits their precocious differentiation into neuromasts. The following experiments were designed to identify signaling pathways that are regulated in interneuromast cells in response to Schwann cell-derived signals.

## Proliferation is the first cellular response in interneuromast cells after abrogation of the ErbB pathway

The identification of interneuromast cell behaviors that are inhibited by ErbB signaling provides clues to which signaling pathways might be regulated by ErbB signaling. To identify the earliest changes in lateral line cell behavior in response to the loss of ErbB signaling we performed time-lapse analyses. We imaged interneuromast cells in Schwann cell-depleted larvae derived from crosses between *Tg(sox10:DNerbb4)* and *Tg(SqET20:gfp)*. *Tg(SqET20:gfp)* larvae express EGFP in neuromast mantle cells and interneuromast cells (*Parinov et al., 2004*). In a 40-hr time-lapse four intercalary neuromasts are formed from interneuromasts cells (*Video 1*). The time-lapse analyses revealed that interneuromast cell proliferation precedes clustering of interneuromast cells. In addition, interneuromast cells are highly motile and migrate into and out of the forming neuromasts. The clusters of interneuromast

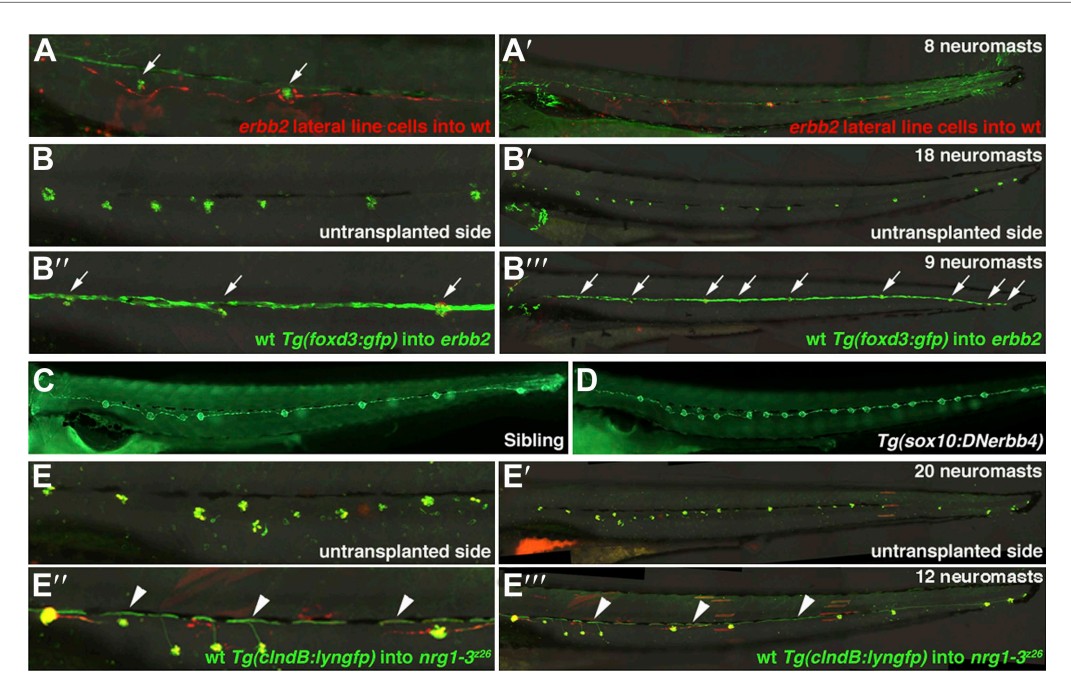

**Figure 2**. Transplantation and transgenic analysis demonstrates that ErbB2 is required within Schwann cells and Nrg1-3 within lateral line neurons to inhibit extra neuromast formation. (**A** and **A'**) Alexa-568 dextran (red) labeled *erbb2* mutant cells were transplanted into *Tg(foxd3:gfp)* (green) wild type fish. (**A**) High magnification view shows *erbb2* interneuromast and mantle cells along the lateral line and around neuromasts (arrows). These *erbb2* interneuromast cells fail to induce extra neuromasts by 4 dpf (**A'**). (**B–B'''**) Alexa-568 dextran (red)/*Tg(foxd3:gfp)* (green) wild type cells were transplanted into *erbb2* mutant host. At 4 dpf neuromast were labeled by DASPEI staining (green). (**B** and **B'**) On the untransplanted side there are no Schwann cells and eighteen neuromasts. (**B''** and **B'''**) On the transplanted side you can see complete migration of wild type Schwann cells in an otherwise *erbb2* mutant fish and rescue of neuromast number (arrows). (**C** and **D**) Dominant negative ErbB receptor expression in neural crest derived cells mimics *erbb* mutant phenotype. (**C**) Control *Tg(SqET20:gfp)* siblings at 4 dpf. (**D**) *Tg(SqET20:gfp)/Tg(sox10:DNerbb4)* showing extra neuromasts. (**E–E''**) Alexa-568 dextran (red)/*Tg(clndB:lyngfp)*(green) wild type cells were transplanted into *nrg1-3^{z26}* mutant host. (**E–E'**) At 4 dpf the untransplanted side has nineteen neuromasts. (**E''** and **E'''**) On the transplanted side there are GFP labeled axons (arrowhead) along the entire length of the lateral line with rescue of neuromast number. All axons cannot be seen because some are obscured underneath pigment cells.

cells continue to proliferate and differentiate into neuromasts as evident by the mature pattern of a ring of *Tg(SqET20:gfp)* positive mantle cells that surrounds GFP-negative sensory hair cells. In contrast, control *Tg(SqET20:gfp)* larvae show no proliferation and little migration of interneuromast cells during the same time period (*Video 2*). In conclusion, the absence of Schwann cells, leads first to interneuromast cell proliferation, followed by an increase in migration and clustering of interneuromast cells that eventually differentiate into sensory hair and support cells.

To be able to correlate cell behavior with gene expression changes (see below), we sought to determine how many hours after ErbB inhibition interneuromast proliferation begins. We added BrdU plus DMSO or AG1478 to *Tg(SqET20:gfp)* larvae at 48 hpf, after Schwann cell migration is completed. After 14 hr of ErbB signaling inhibition there is no significant increase in BrdU incorporation in GFP-positive interneuromast cells (*Figure 3A–B,E*). After 24 hr of treatment we detected a significant increase in BrdU labeling in interneuromast cells (*Figure 3C–E*). Concurrent with the increase in BrdU incorporation, an increase in interneuromast cells is observed after 24 hr of AG1478 treatment (*Figure 3F*). The increase in proliferation begins sometime between 14 and 24 hr post ErbB inhibition.

As ErbB signaling acts cell-autonomously in Schwann cells but proliferation occurs in interneuromast cells, we aimed to identify the signaling pathways that are activated in interneuromast cells when ErbB signaling is inhibited.

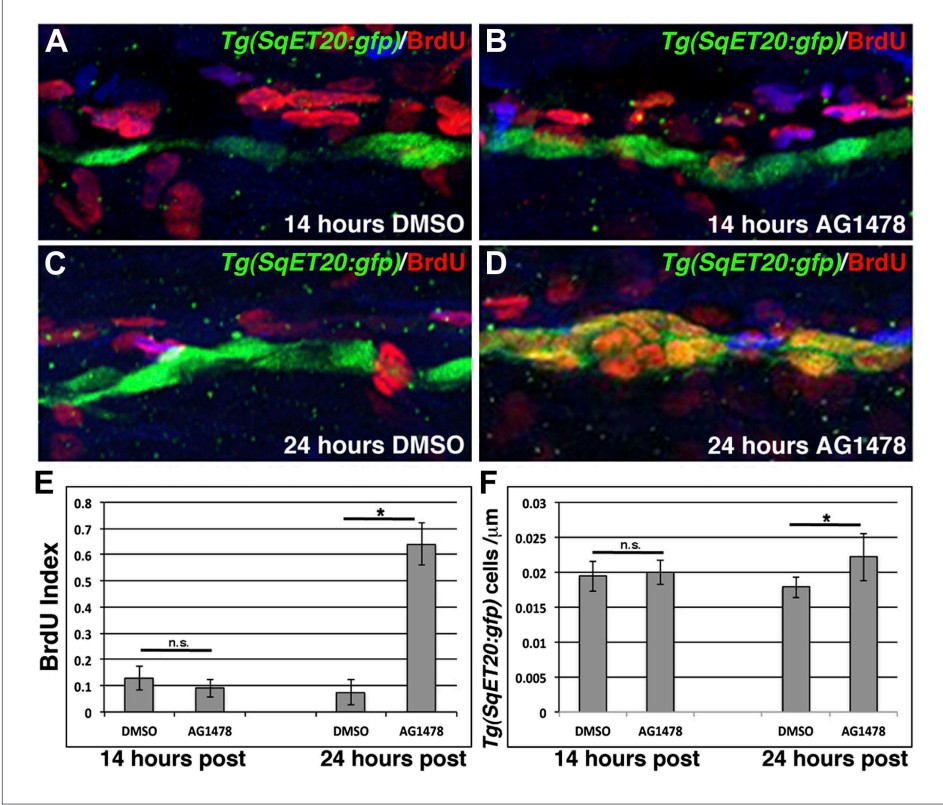

**Figure 3**. ErbB inhibition, after Schwann cell lateral line migration is completed, induces proliferation of interneuromast cells. BrdU plus DMSO or AG1478 was given to *Tg(SqET20:gfp)* fish at 48 hpf then fixed at 14 or 24 hr post treatment. Immunohistochemistry for BrdU (red) and GFP (green) reveals no difference in BrdU incorporation within interneuromast cells between DMSO (**A**) or AG1478 (**B**) 14 hr post treatment. At 24 hr post treatment DMSO (**C**) treated fish show little BrdU incorporation while AG1478 (**D**) treated fish show increased BrdU incorporation and interneuromast cell number. Quantification of both BrdU index (**E**, Student's *t*-test, p=0.18 for 14 hr and p=4.5E$^{-18}$ for 24 hr time point) and interneuromast cell number (**F**, Student's *t*-test, p=0.69 for 14 hr and p=0.003 for 24 hr time point) shows a significant increase with AG1478 only after 24 hr.

## Wnt/β-catenin, Fgf and Notch signaling pathways are upregulated in ErbB/Neuregulin pathway mutants

The Wnt/β-catenin, Fgf and Notch signaling pathways are excellent candidates for being involved in intercalary neuromast formation as they regulate progenitor cell proliferation in several other organs, such as the CNS (*Logan and Nusse, 2004*; *Guillemot and Zimmer, 2011*; *Koch et al., 2013*). In addition, these three pathways play multiple, crucial roles in the primordium of the lateral line (reviewed in *Aman and Piotrowski, 2009*; *Ma and Raible, 2009*; *Chitnis et al., 2012*). Briefly, Wnt/β-catenin and Fgf signaling regulate cell proliferation, while Fgf is also required for neuromast rosette formation and hair cell differentiation (*Aman and Piotrowski, 2008*; *Lecaudey et al., 2008*; *Nechiporuk and Raible, 2008*; *Aman et al., 2011*). Notch signaling regulates sensory hair cell production and primordium cohesion (*Itoh and Chitnis, 2001*; *Matsuda and Chitnis, 2010*). To test if the Wnt/β-catenin, Fgf and Notch pathways are also involved in the extra neuromast phenotype we performed an in situ expression screen on 48 hpf *nrg1-3*$^{z26}$ mutant larvae (*Figure 4*). The Wnt/β-catenin pathway members *wnt10a*, *lef1*, *myca,* and *β-catenin-2* (*ctnnb2*) are upregulated in interneuromast cells in *nrg1-3*$^{z26}$ larvae (*Figure 4A–H*). *lef1* and *ctnnb2* are also expressed at lower levels in interneuromast cells in control animals while *wnt10a* and *myca* show no expression. The expression of *wnt10a* correlates with the differentiation status of intercalary neuromasts. *wnt10a* is expressed in proliferating interneuromast cells but is down regulated in differentiating neuromasts (*Figure 4—figure supplement 1A–D*). The Notch receptor *notch3* and the Notch target gene *her4.1* are expressed in primary neuromasts in

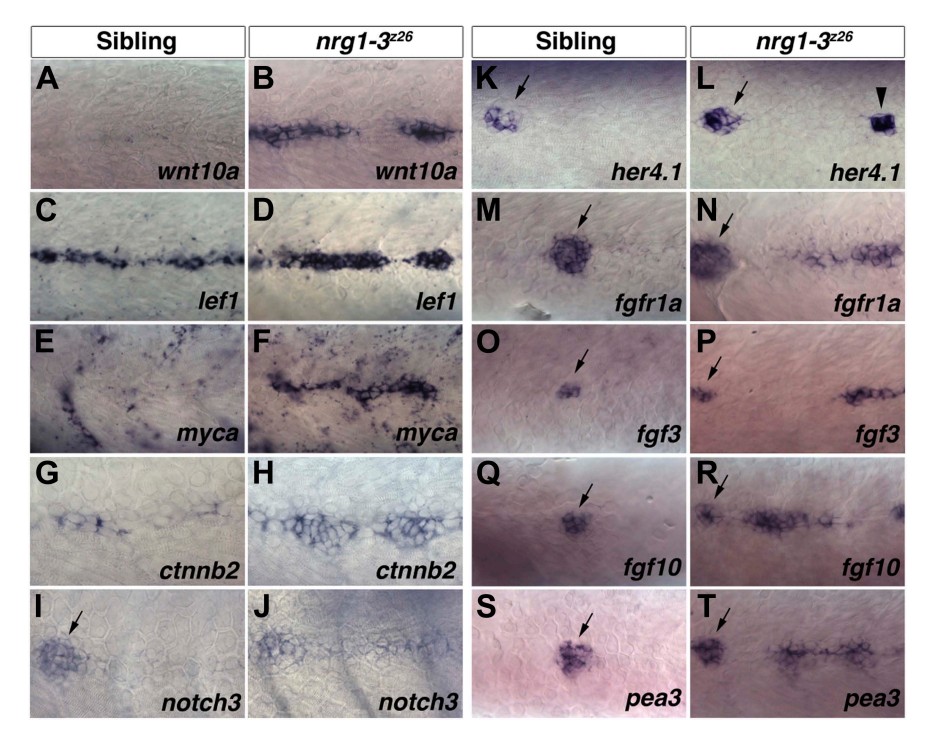

**Figure 4**. Increase in Wnt/β-catenin, Notch and Fgf signaling pathway gene expression in *nrg1-3^{z26}* mutant interneuromast cells. Control siblings and *nrg1-3^{z26}* mutants were processed for in situ hybridization at 48 hpf. A Wnt ligand, *wnt10a*, is not expressed in control interneuromast cells (**A**) but is increased in *nrg1-3^{z26}* (**B**). The Wnt/β-catenin target gene *lef1* is expressed in interneuromast cells in control siblings (**C**) but is greatly increased in *nrg1-3^{z26}* (**D**). An additional Wnt/β-catenin target *myca* shows no expression in interneuromast cells (**E**) but strong expression in clumps of interneuromast cells in *nrg1-3^{z26}* (**F**). *ctnnb2* shows weak expression in control interneuromast cells (**G**) which is upregulated in *nrg1-3^{z26}* (**H**). (**I**) In controls, *notch3* is expressed in primary neuromasts (arrow) but not interneuromast cells. (**J**) *notch3* is upregulated in mutant interneuromast cells. (**K**) In controls, the Notch target *her4.1* is expressed in primary neuromasts (arrow) but not interneuromast cells. (**L**) In mutants, *her4.1* is expressed in primary neuromasts (arrow) but is also increased in discrete clusters of cells (arrowhead). In controls the Fgf pathway genes including the receptor *fgfr1a* (**M**), the two ligands *fgf3* (**O**) and *fgf10* (**Q**) and the Fgf target gene *pea3* (**S**) are all expressed in primary neuromasts (arrow) but not in interneuromast cells. All Fgf pathway genes, *fgfr1a* (**N**), *fgf3* (**P**) *fgf10* (**R**) and *pea3* (**T**), retain expression in primary neuromasts (arrow) but are upregulated in interneuromast cells of *nrg1-3^{z26}*.

The following figure supplements are available for figure 4:

**Figure supplement 1**. *wnt10a* is expressed in proliferating interneuromast cells while *her4.1* is expressed in differentiating and mature neuromasts.

**Figure supplement 2**. *wnt10a*, *fgf3* and *fgf10* expression depends on Wnt/β-catenin signaling.

**Figure supplement 3**. *wnt10a*, *fgf3* and *fgf10* expression are induced within interneuromast cells after Wnt/β-catenin activation.

control animals but not in interneuromast cells (*Figure 4I,K*, arrows). *notch3* is broadly induced in *nrg1-3^{z26}* interneuromast cells (*Figure 4J*). *her4.1* is also induced in *nrg1-3^{z26}* but in a more discrete cluster of cells (*Figure 4L*, arrowhead). *her4.1* is not expressed in proliferating interneuromast cells but is induced as intercalary neuromasts mature (*Figure 4—figure supplement 1E–H*). This suggests that Notch signaling is only active in differentiating neuromasts. Fgf signaling pathway components are also upregulated in interneuromast cells in *nrg1-3^{z26}* mutant larvae (*Figure 4N,P,R,T*). In control embryos, *fgfr1a*, *fgf3*, *fgf10* and the Fgf target *pea3* only show strong expression in primary

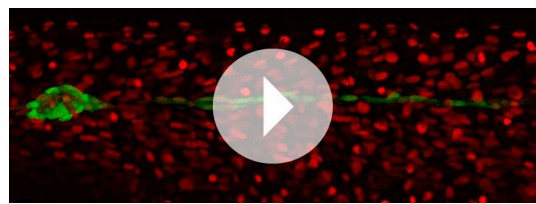

**Video 1**. Time-lapse recording of *Tg(sox10:DNerbb4)/Tg(SqET20:gfp)/Tg(clndB:H2B-mcherry)* during intercalary neuromast formation. The time-lapse runs from 32 to 72 hpf. One frame was taken every 7 min. Four intercalary neuromasts form during this time.

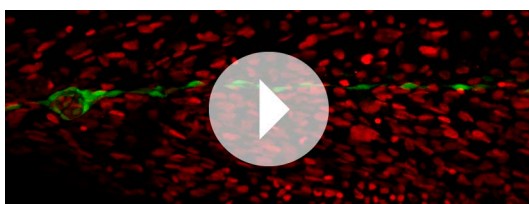

**Video 2**. Time-lapse recording of control *Tg(SqET20:gfp)/(Tg(clndB:H2A-mcherry)* from approximately 48–72 hpf. One frame was taken every 7 min. No interneuromast cell proliferation is seen during this time.

neuromasts but not interneuromast cells (*Figure 4M,O,Q,S*, arrows), suggesting that, similar to Notch signaling, Fgf signaling might be involved in neuromast differentiation.

## Wnt/β-catenin and Fgf pathway members are activated during intercalary neuromast development in wild type larvae

If increased Wnt/β-catenin and Fgf signaling are important for intercalary neuromast formation in Schwann cell-deficient larvae, these pathways should also be upregulated during post-embryonic intercalary neuromast formation in wild type larvae.

To enable us to compare gene expression changes with neuromast formation we quantified neuromasts formed between 2–6 dpf. In wild type larvae two primordia form the majority of the posterior lateral line system (*Ghysen and Dambly-Chaudiere, 2007*). PrimI deposits five to six primary neuromasts between 20–40 hpf. PrimII begins migration at around 40 hpf along the same path as primI (*Sapede et al., 2002*; *Nunez et al., 2009*). We counted the number of neuromasts after alkaline phosphatase staining (*Figure 5D–G*). The total number of neuromasts increases steadily from six to seven neuromasts at 2 dpf to 12 at 6 dpf (*Figure 5A*). Of these 12 neuromasts at 6 dpf, three have been deposited by primII (*Figure 5B,E–G*,

asterisks). PrimII-derived neuromasts are always located dorsally to the primI-derived chain of interneuromast cells. The first clusters of interneuromast cells that will differentiate into intercalary neuromasts appear by 3 dpf (*Figure 5E*, arrowhead). Typically, at least one intercalary neuromast has formed by 4 dpf, with a second formed by 6 dpf (*Figure 5C,F–G*, squares). Therefore, any genes crucial for interneuromast proliferation and differentiation should commence expression between 2–3 dpf.

In situ hybridization experiments for the Wnt/β-catenin targets *lef1* and *ctnnb2* revealed that *lef1* and *ctnnb2* are expressed in deposited interneuromast cells at 2 dpf (*Figure 5H,L*, arrowheads). Unlike *lef1*, *ctnnb2* is also expressed in differentiating neuromasts (*Figure 5M–O*, asterisk). *lef1* and *ctnnb2* are downregulated in interneuromast cells between 3–5 dpf, with the exception of forming clusters of interneuromast cells that will differentiate into intercalary neuromasts (*Figure 5I–K,M–O*, arrowheads). To study *lef1* expression at single cell resolution we photographed the lateral line in *Tg(SqEt20:gfp)* larvae at 4 or 6 dpf and then performed *lef1* in situ hybridization on the same larvae (*Figure 5—figure supplement 1*). These experiments illustrate that *lef1* is expressed only in the few interneuromast cells that begin to proliferate to form clusters (*Figure 5—figure supplement 1A–D*, arrowheads). As *lef1* expression is shut off in mature neuromasts, Wnt/β-catenin signaling is likely involved in the initiation of intercalary neuromast proliferation and not differentiation.

In contrast, the Fgf target *pea3* is never expressed in interneuromast cells but is strongly expressed in differentiated neuromasts from 2–5 dpf (*Figure 5P–S*, asterisks). Discrete clusters of *pea3* expressing interneuromast cells are observed at 5 dpf (*Figure 5S*, arrowhead). The expression analyses shows that Wnt/β-catenin pathway activation coincides with interneuromast cell proliferation and cluster formation, whereas Fgf signaling is initiated later, during the differentiation phase of intercalary neuromast formation.

## Wnt/β-catenin activation occurs prior to Notch and Fgf activation within interneuromast cells after ErbB inhibition

The expression analyses during wild type intercalary neuromast formation suggest that the onset of Wnt/β-catenin expression precedes Fgf and Notch signaling. To determine the temporal dynamics of

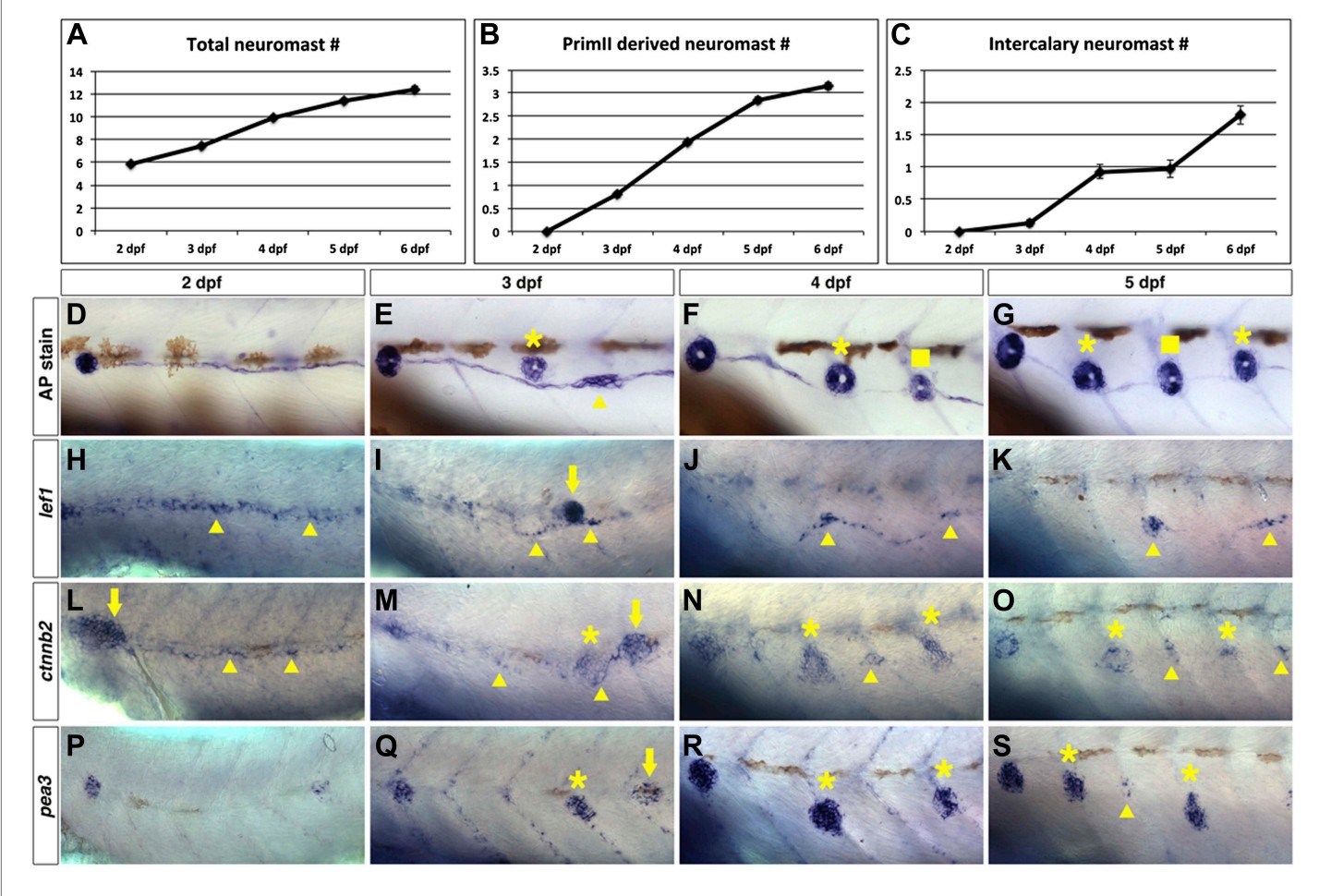

**Figure 5**. Wnt/β-catenin and Fgf signaling target genes are expressed in interneuromast cells during wild type intercalary neuromast formation. To see when intercalary neuromasts first arise we alkaline phosphatase stained wild type zebrafish at 2, 3, 4, 5 and 6 dpf. Quantification of total neuromast number shows a steady increase from 2 to 6 dpf (**A**). Most of the increase comes from primII deposited neuromasts (**B**). There is one intercalary neuro-mast by 4 dpf and two by 6 dpf (**C**). (**D–G**) Alkaline phosphatase staining from 2–5 dpf. The images were taken so that the first primI deposited neuro-mast is always at the left (without a label). Asterisk labels primII-derived neuromasts, the arrowhead labels interneuromast cells and squares label intercalary neuromasts. (**H–K**) *lef1* in situ hybridization from 2–5 dpf. *lef1* is expressed in interneuromast cells at 2 dpf (**H**, arrowhead) and is maintained in clumps of interneuromast cells from 3–5 dpf (**I–K**, arrowheads) that will likely give rise to intercalary neuromasts. The strong cluster of *lef1* expression at 3 dpf is the leading edge of primII (**I**, arrow). *lef1* is not expressed in mature neuromasts. (**L–O**) beta-catenin2 (*ctnnb2*) expression from 2–5 dpf. Similar to *lef1*, *ctnnb2* is expressed in interneuromast cells at 2 dpf (L, arrowhead) and is maintained in clumps of interneuromast cells from 3–5 dpf (**M–O**, arrowhead). Unlike *lef1*, *ctnnb2* is expressed in primary neuromasts (**M–O**, asterisk). (**P–S**) *pea3* expression from 2–5 dpf. (**P**) At 2 dpf *pea3* shows strong expression in primary neuromasts but not in interneuromast cells. (**Q–R**) At 3 and 4 dpf *pea3* shows expression in primII (arrow) and primII derived neuromasts (asterisk) but still no interneuromast cell expression. At 5 dpf *pea3* can be seen in a few cells near somite boundaries (**S**, arrowhead).

The following figure supplements are available for figure 5:

**Figure supplement 1**. *lef1* is expressed in clusters of interneuromast cells before they become intercalary neuromasts.

pathway activations after ErbB inhibition we blocked ErbB signaling with AG1478 at 48 hpf and fixed larvae at 6, 12, 24 and 36 hr post treatment followed by in situ hybridization. We focused our analysis on interneuromast cells between the first and second deposited primary neuromasts. We performed in situ hybridization with the Wnt/β-catenin target *lef1* and a *dgfp* probe for a transgenic Wnt/β-catenin reporter line, *Tg(Tcf/Lef-miniP:dGFP)* that contains 6 copies of a consensus TCF/Lef binding site followed by destabilized EGFP (**Shimizu et al., 2012**). At 6 and 12 hr post treatment, DMSO treated fish show strong expression of the Wnt/β-catenin reporter in primII (**Figure 6A–B**, arrow), and in some interneuromast cells (arrowhead). At 24 hr post DMSO the Wnt/β-catenin

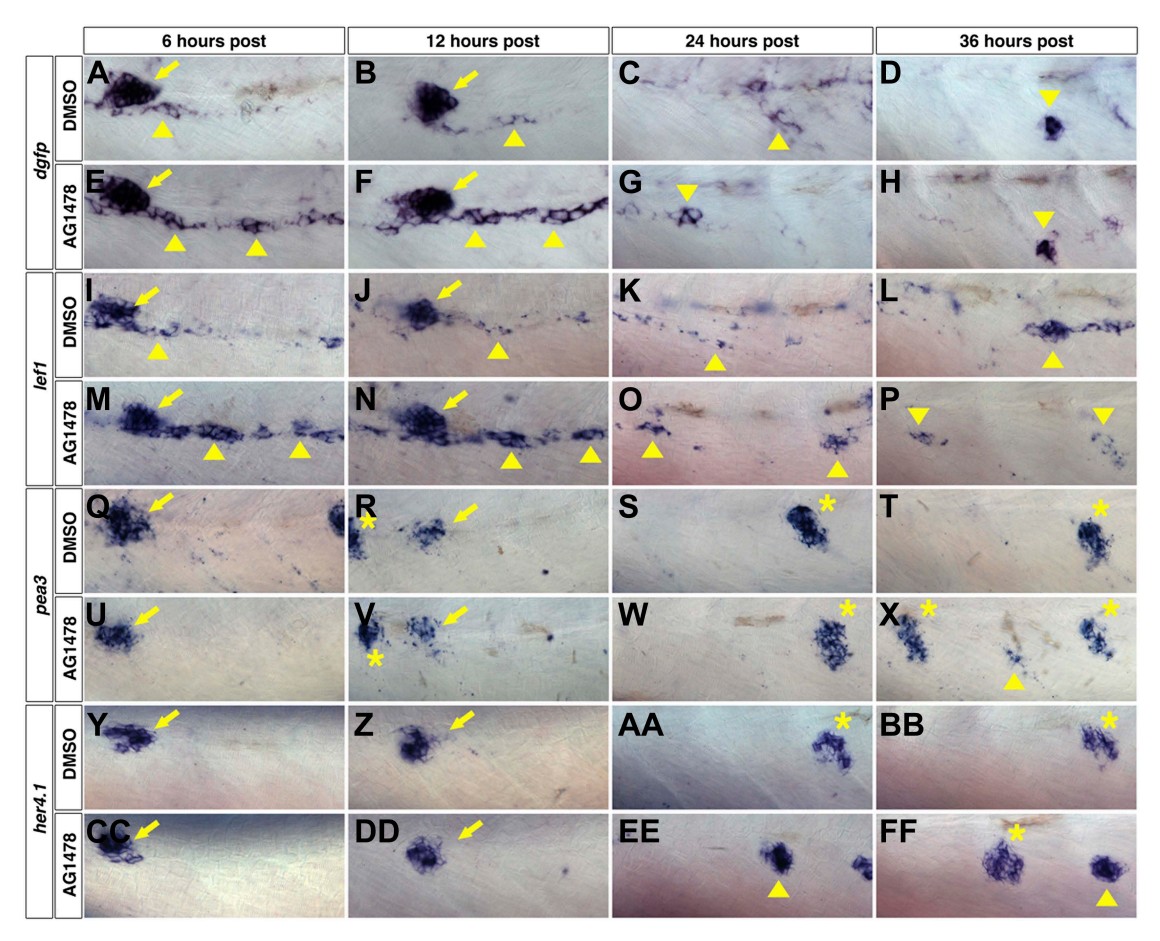

**Figure 6**. Wnt/β-catenin expression precedes Fgf and Notch expression after ErbB inhibition. To determine which signaling pathways are induced first we treated wild type zebrafish starting at 48 hpf with DMSO or AG1478 then fixed at 6, 12, 24 and 36 hr post treatment. All images were taken between the first two primary neuromasts. In DMSO treated Wnt reporter Tg(*Tcf/Lef-miniP:dGFP*) strong expression is seen in primII (arrow) at 6 and 12 hr post treatment but also in some interneuromast cells (**A–B**, arrowhead). By 12 and 36 hr there are only clumps of interneuromast cells expressing the reporter (**C–D**, arrowhead). 6 hr post AG1478 treatment there is a large increase in Wnt reporter expression specifically within interneuromast cells (**E**, arrowhead). This AG1478 induced increased expression is maintained after 12 hr (**F**) but has started to go down by 24 hr (**G**) and is only seen in a few interneuromast cells by 36 hr (**H**). (**I–P**) *lef1* mirrors Wnt reporter expression. In DMSO, *lef1* is expressed in primII (arrow) and few interneuromast cells at 6 (**I**) and 12 hr (**J**). At 24 hr post DMSO *lef1* is maintained in a few interneuromast cells (**K**). At 36 hr post DMSO, *lef1* is seen in clumps of cells likely corresponding to normally developing intercalary neuromasts (**L**, arrowhead). AG1478 induces *lef1* within interneuromast cells at 6 (**M**) and 12 hr (**N**). By 24 (**O**) and 36 hr (**P**) post AG1478 treatment *lef1* is decreased in interneuromast cells compared to the 6 and 12 hr time points, and is maintained in a few clumps of interneuromast cells (arrowhead). (**Q–X**) *pea3* shows a later induction than Wnt target genes. In controls *pea3* is only seen in primII (arrow) or primII deposited neuromasts (asterisk) at all time points tested (**Q–T**). (**U–X**) After AG1478 treatment, *pea3* is maintained in deposited neuromasts (asterisk) and begins to be expressed in interneuromast cells only at 36 hr post treatment (**X**, arrowhead). (**Y–FF**) *her4.1* expression is seen before *pea3* expression. (**Y–BB**) In controls, *her4.1* is seen in primII (arrow) and deposited neuromasts (asterisk). In AG1478 treated larvae *her4.1* is seen in large clusters of cells starting at 24 hr post treatment and can still be seen at 36 hr (**EE–FF**, arrowhead).

reporter is expressed in few interneuromast cells (*Figure 6C*, arrowhead). By 36 hr post DMSO the Wnt/β-catenin reporter is only expressed in clusters of newly forming intercalary neuromasts and is downregulated in other interneuromast cells (*Figure 6D*, arrowhead). This correlates with the clusters of *lef1* expression seen in wild type fish at 3–5 dpf (*Figure 5I–K*). ErbB inhibition induces a large increase of the Wnt/β-catenin reporter expression in interneuromast cells at 6 and 12 hr post treatment (*Figure 6E–F*, arrowheads). By 24 and 36 hr post ErbB inhibition the Wnt/β-catenin reporter expression level has decreased and is only maintained in a few clumps of interneuromast cells (*Figure 6G–H*, arrowhead). The Wnt/β-catenin reporter is not seen in primary neuromasts

(*Wada et al., 2013b*), and is turned off in interneuromast cell clusters as they differentiate into intercalary neuromasts (data not shown). The pattern and timing of *lef1* expression mirrors the Wnt/β-catenin reporter expression, with ErbB inhibition inducing high expression at early stages followed by a gradual decrease (*Figure 6I–P*).

The temporal expression analysis of the Fgf target *pea3* shows that the Fgf pathway is also induced after ErbB inhibition but that the induction happens at a much later time point compared to the Wnt/β-catenin pathway. From 6 to 24 hr post treatment with DMSO or AG1478 *pea3* shows no expression in interneuromast cells, even though it is strongly expressed in primII and in deposited neuromasts (*Figure 6Q–S,U–W*). At 36 hr post ErbB inhibition *pea3* appears in small clusters of interneuromast cells (*Figure 6X*, arrowhead). No *pea3* expression is seen in interneuromast cells of control treated fish at 36 hr (*Figure 6T*). This later induction of *pea3* expression after AG1478 correlates with the later induction seen during wild type intercalary neuromast formation (*Figure 5*).

To test when the Notch pathway becomes active after ErbB inhibition we examined the expression of its target gene *her4.1*. Similar to *pea3*, *her4.1* is only expressed in primII and deposited neuromasts but not in interneuromast cells in control embryos (*Figure 6Y–BB*). After ErbB inhibition *her4.1* is upregulated in large clusters of cells at 24 hr post treatment. The Notch pathway is induced many hours later than the Wnt/β-catenin pathway but before the activation of Fgf signaling (*Figure 6EE–FF*, arrowhead).

## Wnt and Fgf ligands are dependent on Wnt/β-catenin signaling

In the migrating primordium the Fgf ligands *fgf3* and *fgf10* are Wnt/β-catenin targets, whereas it is not known what induces *wnt10a* (*Aman and Piotrowski, 2008*). To test if *wnt10a*, *fgf3* and *fgf10* are Wnt/β-catenin targets during intercalary neuromast formation we treated control or *Tg(sox10:DNerbb4)* larvae with pharmacological inhibitors of Wnt/β-catenin, Fgfr or Notch signaling at 32 hpf then fixed for in situ hybridization at 48 hpf. To block Wnt/β-catenin signaling we used the Axin2 stabilizing drug IWR-1 (*Chen et al., 2009*). We blocked Fgfr or Notch signaling with PD173074 and the γ-secretase inhibitor LY411575, respectively (*Mohammadi et al., 1998*; *Wong et al., 2004*). *wnt10a*, *fgf3* and *fgf10* expression were only inhibited by blocking Wnt/β-catenin, but not Fgfr or Notch signaling (*Figure 4—figure supplement 2*). To test if Wnt/β-catenin signaling induces Wnt or Fgf ligand expression we treated larvae with BIO, a pharmacological inhibitor of the Wnt/β-catenin inhibitor GSK-3 (*Meijer et al., 2003*). *wnt10a* is induced within 6 hr of treatment and expression is maintained up to 24 hr post-treatment (*Figure 4—figure supplement 3G-L*). *fgf3* and *fgf10* are also induced by BIO treatment, but induction takes longer (*Figure 4—figure supplement 3M–X*). These experiments show that Wnt/β-catenin signaling is both necessary and sufficient for expression of *wnt10a*, *fgf3* and *fgf10* within interneuromast cells of Schwann cell deficient larvae.

The expression time course analysis of Wnt/β-catenin, Fgf and Notch target genes demonstrates that the loss of ErbB signaling leads to a fast activation of the Wnt/β-catenin pathway followed by Notch and Fgf pathway activation several hours later. As Wnt/β-catenin signaling precedes and coincides with interneuromast cell proliferation, Wnt/β-catenin signaling might be required for proliferation. At the same time Wnt/β-catenin signaling is required for *fgf* ligand induction. Notch and Fgf signaling are only upregulated in clusters of cells suggesting that they might be playing a later role in hair cell differentiation and rosettogenesis as during early development.

## Wnt/β-catenin signaling is required for extra neuromast formation

To test if Wnt/β-catenin signaling is necessary for extra neuromast formation we employed several methods to block Wnt/β-catenin signaling. First we analyzed larvae mutant for the Wnt/β-catenin signal transducer *lef1*, as they show a decrease in primordium cell proliferation (*Gamba et al., 2010*; *McGraw et al., 2011*; *Valdivia et al., 2011*). During development *lef1* mutant larvae generate intercalary neuromasts, but not as many as control larvae (*Figure 7—figure supplement 1A*). Lack of *lef1* also partially rescued the extra neuromast phenotype induced by ErbB inhibition (*Figure 7—figure supplement 1B–F*). It is likely that intercalary neuromast formation in *lef1* mutant larvae was only incompletely inhibited due to redundancy with three additional TCF family members that are also expressed in the developing lateral line (*McGraw et al., 2011*; *Valdivia et al., 2011*).

To pharmacologically block Wnt/β-catenin signaling we soaked the larvae in Tankyrase inhibitors. We treated 32 hpf wild type and *Tg(sox10:DNerbb4)* larvae with IWR-1 or XAV939 (*Huang et al., 2009*). Larvae were soaked in the inhibitors for 24 hr and then transferred to embryo medium. At 3 dpf

we counted the number of alkaline phosphatase stained neuromasts (*Figure 7C–H*), up to somite 14 because both inhibitors induced stalling of primI migration (data not shown and *Matsuda et al., 2013*). Wnt/β-catenin inhibition induced a decrease in neuromast number in wild type siblings due to an effect on primII deposition (*Figure 7A*, blue bars). Importantly, both Wnt/β-catenin pathway inhibitors significantly decreased extra neuromast formation compared to DMSO in *Tg(sox10:DNerbb4)* larvae (*Figure 7A*, red bars).

As a third method to inhibit Wnt/β-catenin signaling we heat shock induced Dkk1b (*Stoick-Cooper et al., 2007*), a secreted antagonist of Wnt/β-catenin signaling that blocks Wnt/β-catenin dependent gene expression and cell proliferation in the lateral line primordium (*Aman and Piotrowski, 2008*; *Aman et al., 2011*). *Tg(hsp70l:dkk1b)* zebrafish were crossed to *Tg(sox10:DNerbb4)* and embryos were heat shocked at 32 hpf. Alkaline phosphatase staining at 3 dpf shows a clear reduction in intercalary neuromasts in *Tg(hsp70l:dkk1b)/Tg(sox10:DNerbb4)* compared to *Tg(sox10:DNerbb4)* (*Figure 7I–L*). Quantification of neuromast number up to somite fourteen verified a complete absence of extra neuromast formation induced by dominant-negative ErbB when Wnt/β-catenin signaling is inhibited by Dkk1b overexpression (*Figure 7B*). These different methods of blocking Wnt/β-catenin signaling all show that this pathway is necessary for intercalary neuromast formation in the absence of ErbB signaling.

## Wnt/β-catenin signaling is required for interneuromast cell proliferation

The lack of extra neuromast formation after Wnt/β-catenin inhibition could be due to a decrease in cell proliferation or a lack of differentiation. Because ErbB inhibition induces an increase in Wnt/β-catenin pathway expression prior to the increase in interneuromast proliferation (*Figures 3 and 6*), we hypothesized that the inability to induce extra neuromasts after Wnt/β-catenin inhibition is due to a lack of proliferation. We performed BrdU incorporation analyses on *Tg(hsp70l:dkk1b)/Tg(sox10:DNerbb4)/Tg(SqET20:gfp)* larvae. Larvae were heat shocked at 32 hpf, then raised in BrdU solution until 48 hpf and fixed. Immunostaining for BrdU and GFP shows a significant increase in double labeling in *Tg(sox10:DNerbb4)* compared to control siblings (*Figure 8A–B,E*). Such increase in BrdU incorporation is not observed in *Tg(hsp70l:dkk1b)/Tg(sox10:DNerbb4)* double transgenic larvae (*Figure 8D–E*). *Tg(hsp70l:dkk1b)* larvae show no significant difference in the BrdU index compared to control larvae (*Figure 8C,E*). Quantification of the number of *Tg(SqET20:gfp)*-positive interneuromast cells shows a significant decrease in *Tg(hsp70l:dkk1b)/Tg(sox10:DNerbb4)* larvae compared to *Tg(sox10:DNerbb4)* larvae (*Figure 8F*). These experiments demonstrate that inhibition of Wnt/β-catenin prevents the formation of intercalary neuromasts in ErbB signaling deficient larvae by inhibiting proliferation of progenitor cells.

## Increased Wnt/β-catenin signaling is sufficient to induce interneuromast cell proliferation

To examine if Wnt/β-catenin signaling is sufficient to induce proliferation we treated 48 hpf larvae with BIO. In DMSO treated larvae *lef1* is expressed in primII and weakly in a few interneuromast cells (*Figure 9A*, arrowhead). BIO treatment induced expression of *lef1* in neuromasts and interneuromast cells (*Figure 9B*, *Figure 4—figure supplement 3*). To study proliferation in interneuromast cells *Tg(SqET20:gfp)* larvae were treated with DMSO or BIO in the presence of BrdU. Similarly to ErbB inhibition, BIO induces an increase in proliferation and the number of interneuromast cells 24 hr post treatment (*Figure 9C–J*). BIO treatment does not result in a reduction of Schwann cells along the lateral line (*Figure 9—figure supplement 1*). The BIO-induced increase in interneuromast cell proliferation did not result in extra neuromasts, due to a strong increase in cell death in interneuromast cells after prolonged BIO treatment (data not shown). We also transplanted *apcmcr* mutant lateral line cells into wild type hosts and similarly observed that these clones did not survive more than 48 hpf (data not shown). This increase in cell death was only seen in interneuromast cells and not in primary neuromasts, suggesting that interneuromast cells are particularly sensitive to the levels of Wnt/β-catenin signaling. Combined, our experiments demonstrate that Wnt/β-catenin signaling is sufficient and necessary for inducing interneuromast cell proliferation and is absolutely required for the extra neuromast formation that occurs in the absence of Schwann cells.

## Fgf signaling is required for intercalary neuromast formation

Fgf signaling is upregulated in interneuromast cells in *nrg1-3z26* mutant larvae (*Figure 4*). To test if Fgf signaling is also required for intercalary neuromast formation we used both genetic and pharmacological methods to block Fgfr signaling. We added the Fgfr inhibitors SU5402 or PD173074

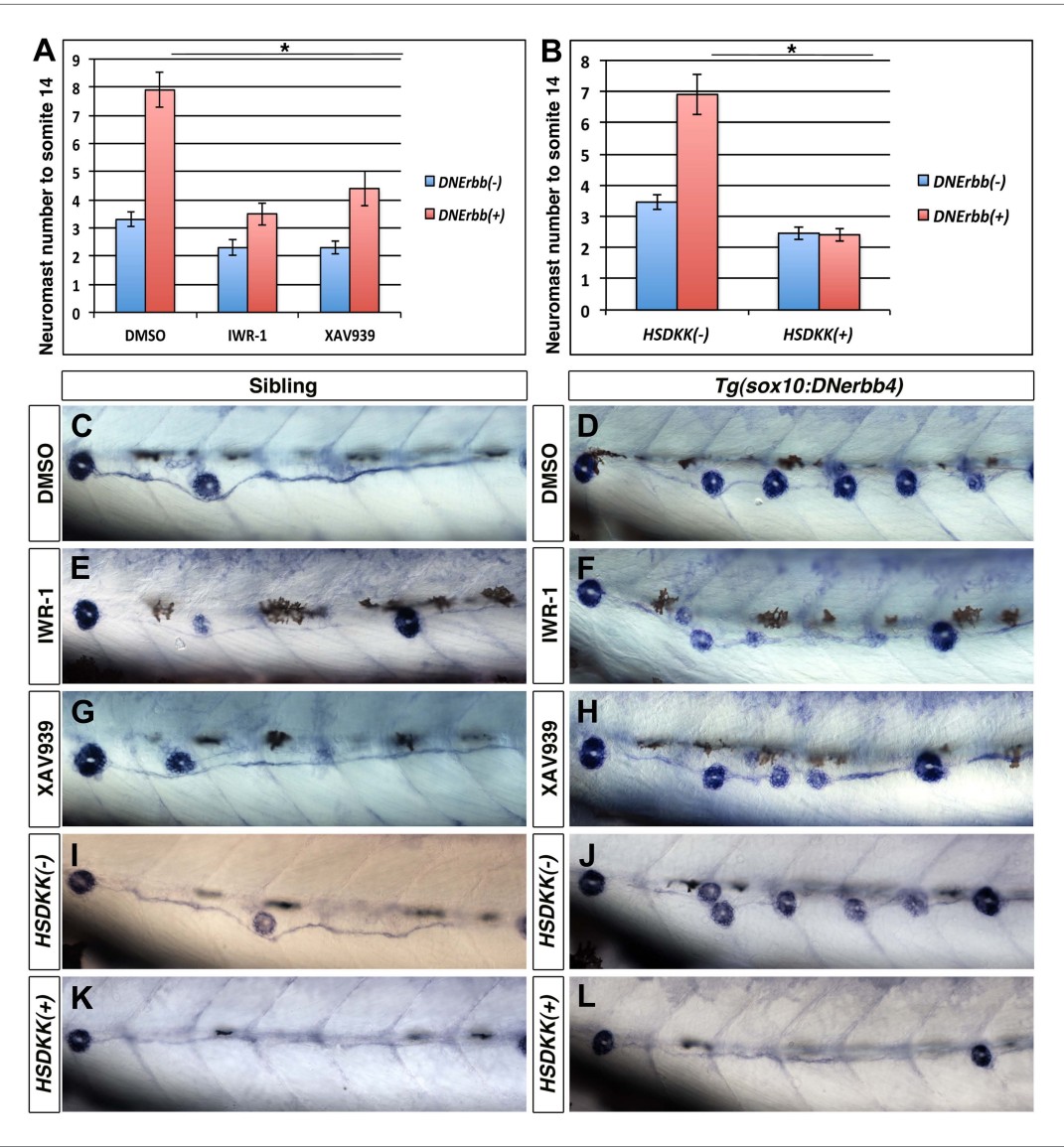

**Figure 7**. Wnt/β-catenin signaling is required for extra neuromast formation in the absence of ErbB signaling. To block Wnt/β-catenin signaling wild type or *Tg(sox10:DNerbb4)* fish were treated with two different inhibitors IWR-1 or XAV939 for 24 hr starting at 32 hpf. Neuromast number up to somite 14 was counted at 3 dpf (**A**). Compared to DMSO, both IWR-1 and XAV939 significantly inhibited neuromast formation in *Tg(sox10:DNerbb4)* (**A**, red bars, One-way ANOVA with Tukey pairwise comparison, p≤0.05). Representative images of alkaline phosphatase stained control siblings treated with DMSO (**C**), IWR-1 (**E**) or XAV939 (**G**) or *Tg(sox10:DNerbb4)* treated with DMSO (**D**), IWR-1 (**F**) or XAV939 (**H**). (**B**) Neuromast counts at 3 dpf of control, *Tg(sox10:DNerbb4)*, *Tg(hsp70l:dkk1b)* or *Tg(sox10:DNerbb4)/Tg(hsp70l:dkk1b)* after heat shock at 32 hpf. *Tg(sox10:DNerbb4)/Tg(hsp70l:dkk1b)* double transgenics show a complete loss of extra neuromast formation seen in *Tg(sox10:DNerbb4)* (**B**, Student's *t*-test, p=2.4E$^{-16}$). Representative images of alkaline phosphatase stained sibling (**I**), *Tg(sox10:DNerbb4)* (**J**), *Tg(hsp70l:dkk1b)* (**K**) or *Tg(sox10:DNerbb4)/Tg(hsp70l:dkk1b)* (**L**) at 3 dpf. The first deposited neuromast is to the left for all alkaline phosphatase images.

The following figure supplements are available for figure 7:

**Figure supplement 1**. *lef1* mutants have decreased intercalary neuromast formation in the absence of ErbB signaling.

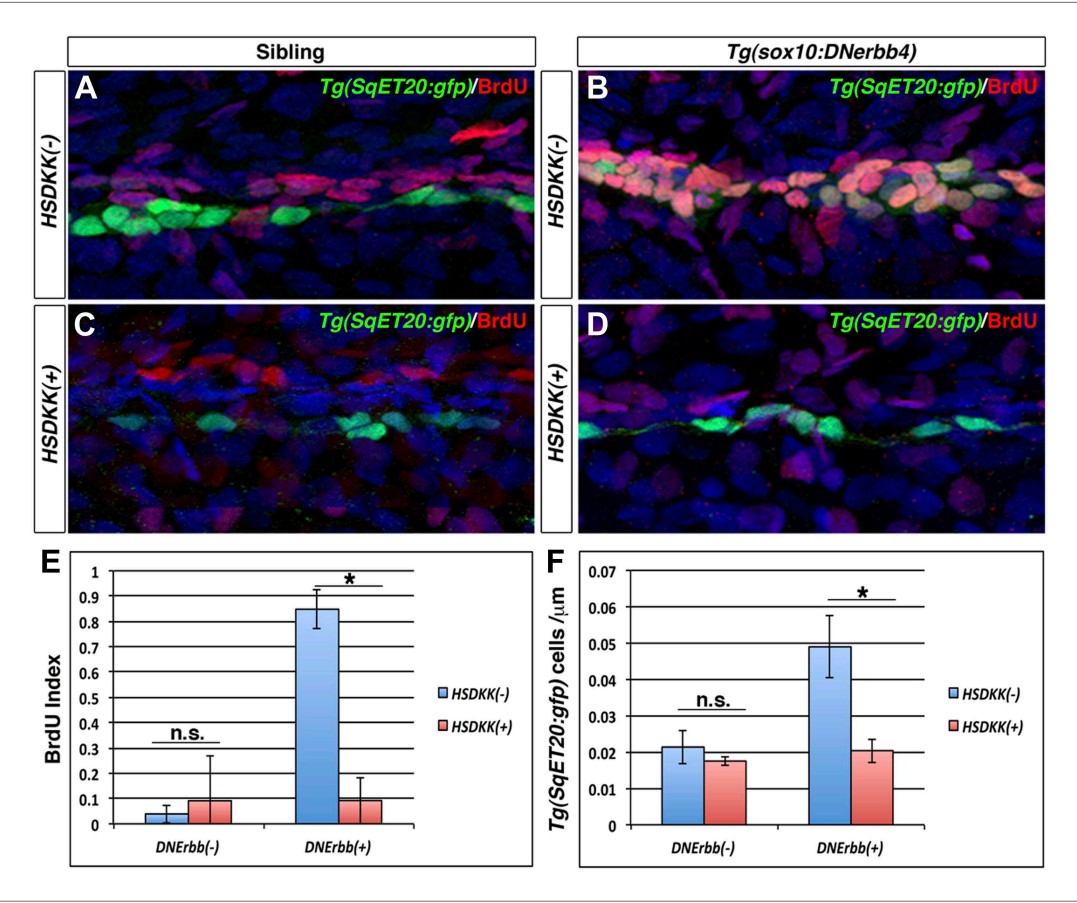

**Figure 8**. Wnt/β-catenin signaling is required for proliferation of interneuromast cells in the absence of ErbB signaling. Control sibling, *Tg(sox10:DNerbb4)*, *Tg(hsp70l:dkk1b)* or *Tg(sox10:DNerbb4)/Tg(hsp70l:dkk1b)*, all with *Tg(SqET20:gfp)* in the background, were heat shocked at 32 hpf then placed in BrdU solution. Fish were fixed at 48 hpf and processed for BrdU and GFP immunohistochemistry (**A**–**D**). Compared to control siblings (**A**), *Tg(sox10:DNerbb4)* (**B**) show a strong increase in BrdU incorporation into *Tg(SqET20:gfp)* positive interneuromast cells. BrdU incorporation in *Tg(sox10:DNerbb4)* is blocked by over expression of Dkk1b (**D**). (**E**) Quantification of BrdU index shows no difference between control and *Tg(hsp70l:dkk1b)* (Student's *t*-test, p=0.6) but a large decrease between *Tg(sox10:DNerbb4)* and *Tg(sox10:DNerbb4)/Tg(hsp70l:dkk1b)* (Student's *t*-test, p=3.4 E$^{-8}$). (**F**) Quantification of *Tg(SqET20:gfp)* cell number again shows no difference between control and *Tg(hsp70l:dkk1b)* (Student's *t*-test, p=0.14) but a large decrease between *Tg(sox10:DNerbb4)* and *Tg(sox10:DNerbb4)/Tg(hsp70l:dkk1b)* (Student's *t*-test, p=1 E$^{-4}$).

to 32 hpf *Tg(SqET20:gfp)* or *Tg(SqET20:gfp)/Tg(sox10:DNerbb4)* larvae and observed neuromasts at 3 dpf. DMSO treated *Tg(sox10:DNerbb4)* larvae show formation of intercalary neuromasts (*Figure 10B*, arrowheads). In contrast, intercalary neuromast formation is blocked after Fgfr inhibition in *Tg(sox10:DNerbb4)* (*Figure 10D,F*). In Fgfr inhibitor treated *Tg(sox10:DNerbb4)* larvae a thickening of the chain of interneuromast cells occurs that is not observed in Fgfr inhibited siblings (*Figure 10C–F*, arrows). However, this increase in interneuromast cells does not lead to the formation of rosettes or differentiated neuromasts. Quantification of neuromast numbers revealed a significant reduction in Fgfr inhibitor treated *Tg(sox10:DNerbb4)* larvae (*Figure 10G*, red bars). Fgfr is required for maintenance of rosettes in the primordium (*Nechiporuk and Raible, 2008*). SU5402 and to a lesser extent PD173074, also induced loss of the rosette shape of primary neuromasts, decreasing the number of neuromasts in treated control siblings (*Figure 10G*, blue bars). As another means to inhibit Fgf signaling, we heat shock induced dominant negative Fgfr1 expression in 32 hpf *Tg(sox10:DNerbb4)* and sibling larvae. Dominant negative Fgfr1 also inhibited extra neuromast formation in *Tg(sox10:DNerbb4)* larvae by 3 dpf (*Figure 10H*). The observation that Fgfr inhibited *Tg(sox10:DNerbb4)* larvae still form

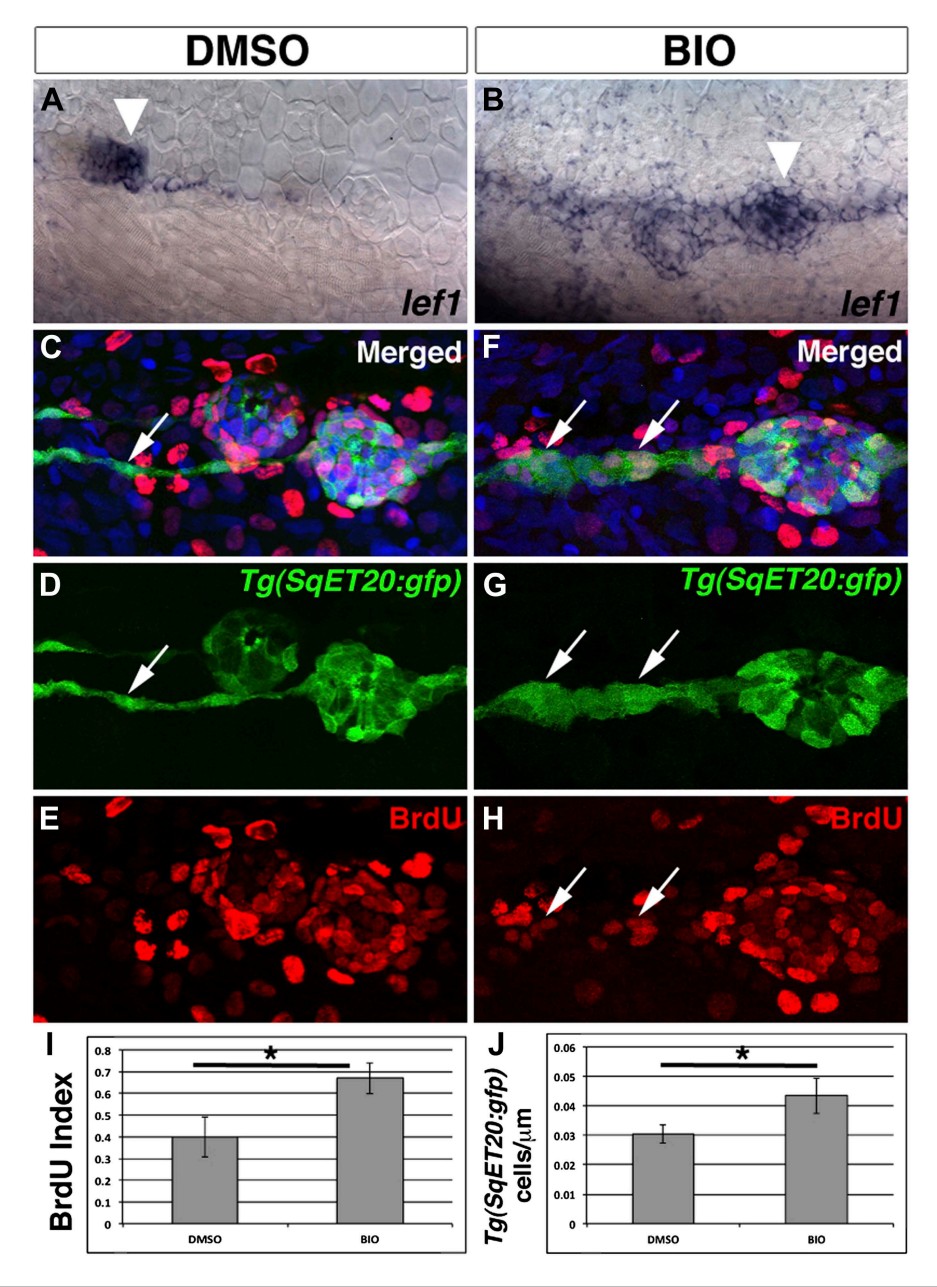

**Figure 9**. Pharmacological activation of Wnt/β-catenin signaling is sufficient to induce interneuromast cell proliferation. To verify BIO induces Wnt/β-catenin signaling we treated 48 hpf zebrafish with DMSO or BIO for 6 hr and then fixed and stained for *lef1* expression. Compared to DMSO treated fish, BIO induces expression of the Wnt/β-catenin target *lef1* in neuromasts and interneuromast cells (**A–B**). The large cluster of cells that are labeled in both (**A**) and (**B**) is primII (arrowhead). To measure proliferation *Tg(SqET20:gfp)* fish were treated with BrdU plus DMSO or BIO at 48 hpf and fixed at 72 hpf. (**C–E**) DMSO treated interneuromast cells (arrow) show single chain morphology with rare BrdU incorporation. BrdU incorporation is strong in primary neuromasts. (**F–H**) BIO treated fish show increased interneuromast cells with BrdU incorporation (arrows). Quantification of BrdU index (**I**, Student's t-test, p=0.0003) and *Tg(SqET20:gfp)* positive cell number (**J**, Student's t-test, p=2.75 E$^{-5}$) shows a significant increase in both after BIO treatment.

The following figure supplements are available for figure 9:

**Figure supplement 1**. Schwann cells are still present after BIO treatment.

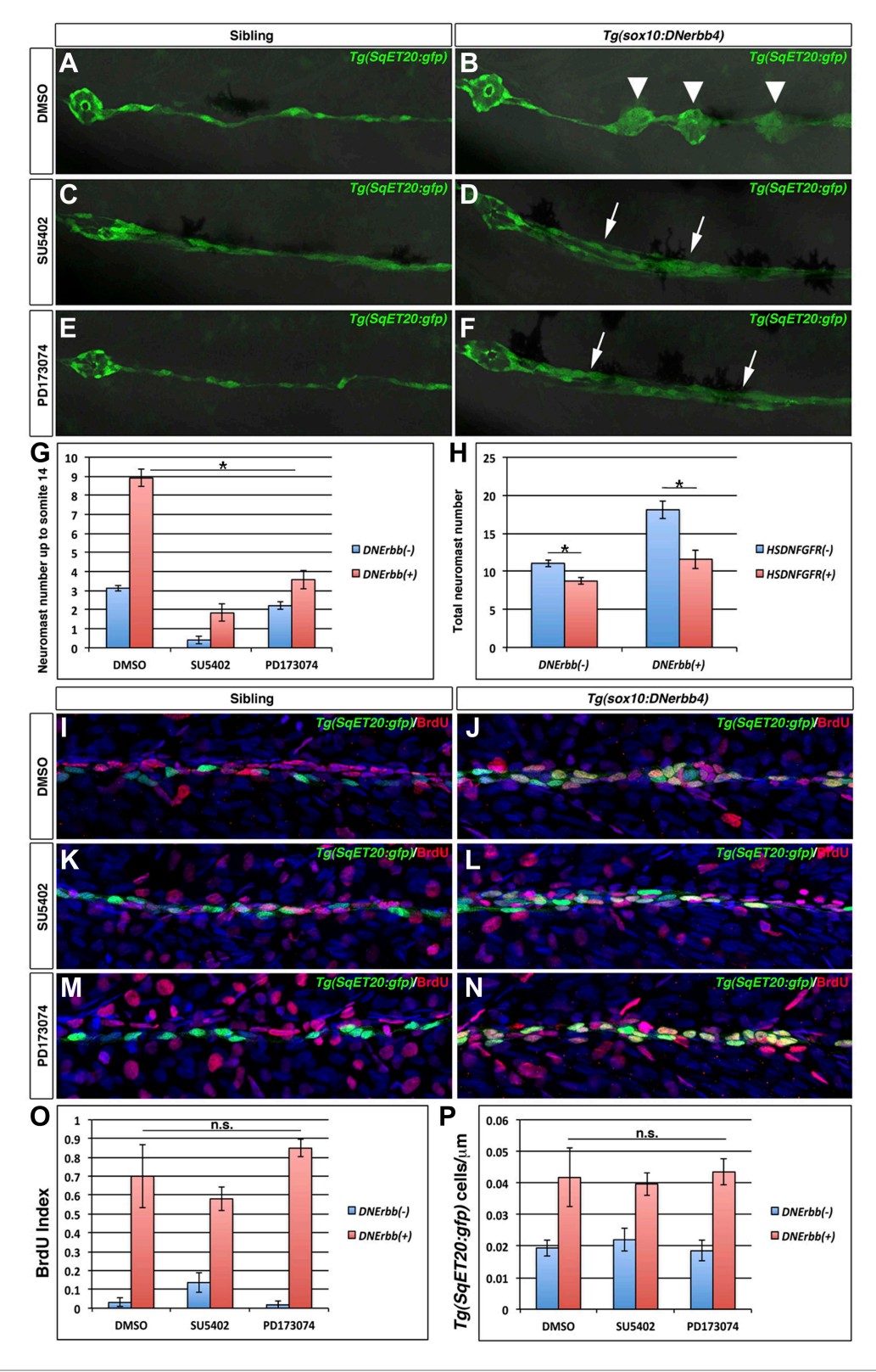

**Figure 10**. Fgf signaling is required for neuromast formation but not interneuromast proliferation in the absence of ErbB signaling. Control *Tg(SqET20:gfp)* or *Tg(sox10:DNerbb4)/Tg(SqET20:gfp)* larvae were treated for 24 hr starting at 32 hpf with DMSO, SU5402 or PD173074 then allowed to develop until 3 dpf and imaged (**A–F**). The first
*Figure 10. Continued on next page*

*Figure 10. Continued*

primary neuromast is to the left in all images. Control siblings (**A**) show no intercalary neuromasts, while *Tg(sox10:DNerbb4)* have several (**B**, arrowhead). *Tg(sox10:DNerbb4)* treated with SU5402 (**D**) or PD173074 (**F**) show clumps of interneuromast cells (arrows) compared to sibling SU5402 (**C**) or PD173074 (**E**) treated but no intercalary neuromasts. (**G**) Quantification of neuromast number up to somite 14 shows a decrease when control siblings are treated with SU5402 or PD173074 (blue bars, One-way ANOVA with Tukey pairwise comparison, p≤0.05). (**G**) Extra neuromast formation is inhibited when *Tg(sox10:DNerbb4)* fish are treated with SU5402 or PD173074 (red bars, one-way ANOVA with Tukey pairwise comparison, p≤0.05). (**H**) *Tg(sox10:DNerbb4)* were crossed to *Tg(hsp70l:dnfgfr1-EGFP)* and larvae were heat shocked at 32 hpf then allowed to grow to 3 dpf. DNFgfr1 reduces neuromasts slightly in control siblings (blue bars, Student's *t*-test, p=7.1 E$^{-9}$) and completely blocks extra neuromast formation in *Tg(sox10:DNerbb4)* (red bars, Student's *t*-test, p=1.5 E$^{-7}$). To measure proliferation, control *Tg(SqET20:gfp)* or *Tg(sox10:DNerbb4)/Tg(SqET20:gfp)* larvae were treated with BrdU plus DMSO, SU5402 or PD173074 at 32 hpf then fixed at 48 hpf. Fish were then processed for BrdU and GFP immunohistochemistry (**I–N**). All *Tg(sox10:DNerbb4)/Tg(SqET20:gfp)* treated fish (**J**, **L** and **N**) show higher BrdU incorporation when compared to control *Tg(SqET20:gfp)* siblings (**I**, **K** and **M**). (**O**) Quantification of BrdU index shows no significant difference between DMSO, SU5402 or PD173074 treated *Tg(sox10:DNerbb4)* larvae (red bars, one-way ANOVA with Tukey pairwise comparison, p≥0.1). (**P**) Quantification of *Tg(SqET20:gfp)* cell number shows no difference in *Tg(sox10:DNerbb4)* larvae treated with DMSO, SU5402 or PD173074 (red bars, one-way ANOVA with Tukey pairwise comparison, p≥0.9).

The following figure supplements are available for figure 10:

**Figure supplement 1**. Posterior lateral line ganglion neurons are affected differently by SU5042 or PD173074 treatment.

---

large clusters of interneuromast cells that fail to differentiate into neuromasts, shows that Fgf signaling is crucial for interneuromast differentiation and rosette formation, but not proliferation.

## Fgf signaling is not required for interneuromast cell proliferation but for differentiation

To test if Fgf signaling is required for proliferation we performed BrdU analyses of Fgfr inhibitor treated *Tg(SqET20:gfp)* and *Tg(sox10:DNerbb4)/Tg(SqET20:gfp)* larvae. BrdU plus DMSO, SU5402 or PD173074 were added at 32 hpf, and the fish were fixed and stained at 48 hpf. Non-transgenic siblings treated with DMSO or Fgfr inhibitors display a single chain of interneuromast cells with rare BrdU positive cells (*Figure 10I,K,M*). SU5402 treated wild type larvae show an increase in BrdU incorporation compared to DMSO treated larvae. The increase is significant by a Student's *t*-test but not by an ANOVA analysis that compares all groups. PD173074 treated wild type fish show no significant change in BrdU incorporation compared to DMSO (*Figure 10O*, blue bars). When characterizing SU5402 and PD173074 treated fish we noticed an effect on the posterior lateral ganglion (pllg) size. Quantification of the number of cells in the pllg showed a decrease induced by SU5402 but an increase induced by PD173074 (*Figure 10—figure supplement 1*). The increase in the posterior lateral line ganglion size after PD173074 treatment is similar to recent results that showed that Fgf inhibition resulted in a larger statoacoustic ganglion (*Vemaraju et al., 2012*). Possibly, the two inhibitors affect a different set of Fgf receptors or pathways. Irrespective of their opposing effect on ganglia size, both SU5402 and PD173074 treated *Tg(sox10:DNerbb4)* transgenic larvae show BrdU positive clusters of interneuromast cells (*Figure 10J,L,N*). Likewise, quantification of BrdU indices and interneuromast cell number indicates that neither inhibitor significantly decreases the proportion of proliferating cells or cell number in *Tg(sox10:DNerbb4)* larvae (*Figure 10O–P*, red bars). These results confirm that Fgf signaling is not required for the increase in interneuromast cell proliferation in Schwann cell deficient zebrafish.

In addition to rosette formation, Fgf signaling also regulates *atoh1a*, which is crucial for sensory hair cell differentiation in the mouse and zebrafish ear and lateral line (*Bermingham et al., 1999*; *Sarrazin et al., 2006*; *Millimaki et al., 2007*; *Lecaudey et al., 2008*; *Nechiporuk and Raible, 2008*). We tested if Fgf signaling plays similar roles in postembryonic and precocious intercalary neuromast formation in wild type and Schwann cell deficient larvae, respectively. We examined the expression of the Fgf targets *pea3* and *atoh1a* after DMSO or PD173074 treatment from 32 to 48 hpf in wild type and

*Tg(sox10:DNerbb4)* larvae (**Figure 11**). In DMSO treated larvae *pea3* and *atoh1a* are only expressed in primary neuromasts but not in interneuromast cells (**Figure 11A,E**, inset). After Fgfr inhibition the expression of *pea3* and *atoh1a* are downregulated in primary neuromasts (**Figure 11B,F**, inset). *pea3* and *atoh1a* are strongly upregulated in precociously differentiating interneuromast cells of DMSO treated *Tg(sox10:DNerbb4)* larvae (**Figure 11C,G**). However, this induction of *pea3* and *atoh1a* in *Tg(sox10:DNerbb4)* larvae is blocked by Fgfr inhibition (**Figure 11D,H**). Therefore, Fgf signaling has the same function in rosette formation and hair cell differentiation in postembryonic intercalary neuromast formation, as in primary neuromast formation during primordium migration.

## Discussion

### ErbB signaling regulates interneuromast cell proliferation by activating a Wnt/β-catenin inhibitor

Interneuromast cells are a latent lateral line progenitor cell population capable of giving rise to all cell types of the neuromast. We uncovered a novel mechanism whereby Schwann cells constitute part of an inhibitory niche, regulating interneuromast cell proliferation and differentiation through non-cell-autonomous regulation of Wnt/β-catenin and Fgf signaling downstream of ErbB signaling (see model **Figure 12**).

ErbB receptor expressing Schwann cells co-migrate with the primordium due to the presence of Nrg1-3 ligand in lateral line axons. As interneuromast cells are deposited by the primordium, they are in close contact with Schwann cells. Schwann cells keep interneuromast cells in a quiescent state via the action of an unknown inhibitor (**Figure 12A**). After deposition, interneuromast cells migrate ventrally away from the midline, which coincides with the commencement of proliferation in interneuromast cells (**Figure 12B**). Simultaneously, Schwann cells migrate medially, crossing the basement membrane (**Raphael et al., 2010**). Both of these steps likely contribute to the release of the Schwann cell inhibitory signal during normal development. The release of this inhibitory signal causes the upregulation of Wnt/β-catenin signaling leading to interneuromast proliferation (**Figure 12C**). Eventually, Notch and Fgf signaling is initiated inducing rosette formation and differentiation.

In zebrafish, ErbB signaling serves several functions during the lifetime of a Schwann cell (**Lyons et al., 2005**; **Raphael et al., 2011**). Our ErbB inhibitor treatment, after Schwann cell migration is complete, suggests that ErbB signaling plays an additional role in Schwann cells by regulating an inhibitor

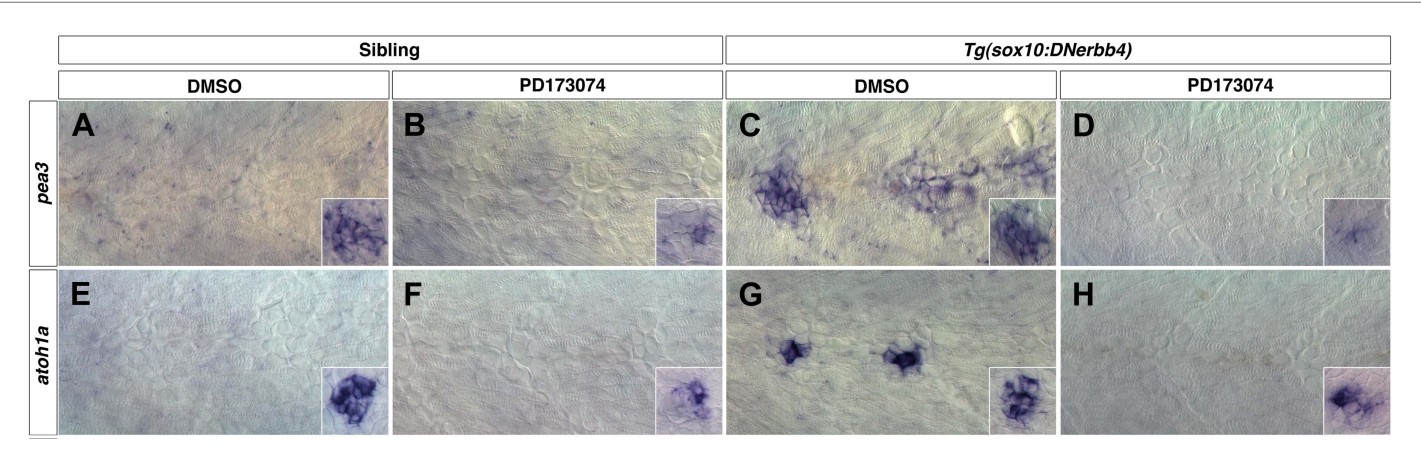

**Figure 11**. Fgf signaling is required for neuromast differentiation. Control or *Tg(sox10:DNerbb4)* siblings were treated with DMSO or PD173074 at 32 hpf then fixed at 48 hpf. To verify that Fgf signaling was blocked, we performed in situ hybridazation for the Fgf target *pea3* (**A–D**). (**A**) In controls treated with DMSO, *pea3* is not expressed in interneuromast cells but is expressed in neuromasts (inset). (**B**) Control siblings treated with PD173074 show downregulation of *pea3* in neuromasts (inset). (**C**) As shown for *nrg1-3*[26], *Tg(sox10:DNerbb4)* have an upregulation of *pea3* within interneuromast cells. (**D**) This upregulation of *pea3* is blocked by PD173074, illustrating that Fgfr signaling is inhibited. (**E**) In controls, *atoh1a* is only expressed in neuromasts (inset). (**F**) PD173074 decreases *atoh1a* in primary neuromasts (inset). (**G**) *atoh1a* is upregulated in differentiating interneuromast cells in *Tg(sox10:DNerbb4)*. (**H**) This upregulation of *atoh1a* in interneuromast cells is completely blocked by PD173074, while some expression is still retained in primary neuromasts (inset).

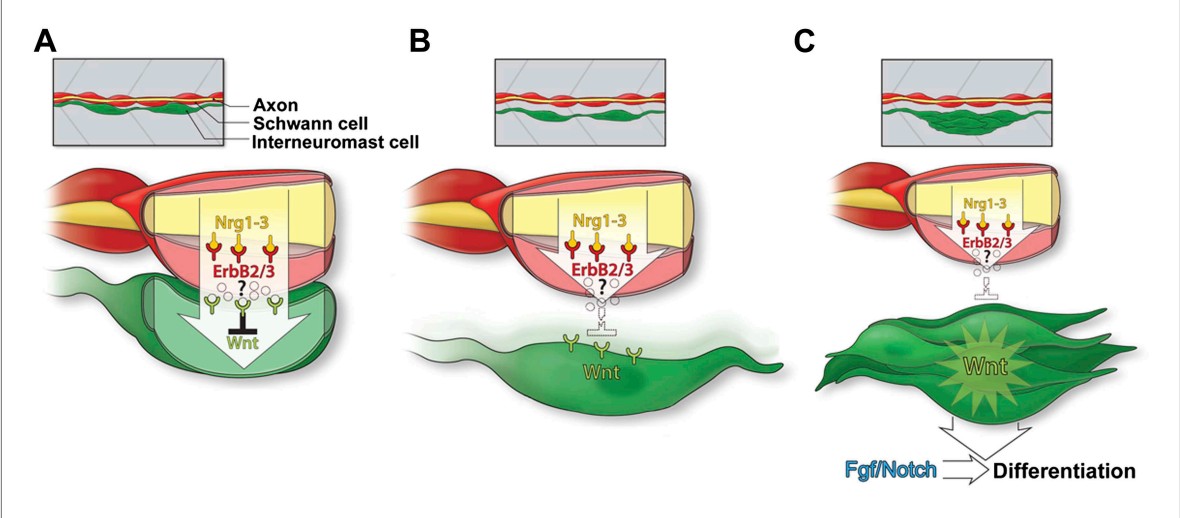

**Figure 12**. Model of Schwann cell inhibition of intercalary neuromast formation. (**A**) Schwann cells (red) co-migrate with lateral line axons (yellow) and interneuromast cells (green). Nrg1-3 present on the axon induces Schwann cell migration and proliferation through activation of ErbB2/3B. As interneuromast cells are deposited they remain in close proximity to Schwann cells. This interaction induces inhibition of Wnt/β-catenin signaling by an unknown mechanism. (**B**) As interneuromast cells migrate ventrally away from Schwann cells, this Wnt/β-catenin inhibition is released. (**C**) Release of inhibition leads to increased Wnt/β-catenin signaling and proliferation followed by Fgf signaling and differentiation.

of Wnt/β-catenin signaling in interneuromast cells. As pharmacological inhibition of ErbB signaling also causes a reduction in Schwann cell proliferation, it is possible that this secondarily affects interneuromast proliferation via reduction of the number of Schwann cells. However, pharmacological inhibition of ErbB signaling induces an increase in Wnt/β-catenin signaling before we observe a decrease in Schwann cell number (*Figure 6* and *Figure 1—figure supplement 4*), suggesting that ErbB signaling directly regulates the expression or activity of the Wnt/β-catenin inhibitor. The inhibitor could either be expressed by Schwann or interneuromast cells.

The list of potential candidates for the Wnt/β-catenin pathway inhibitor is growing (*Cruciat and Niehrs, 2013*). The Wnt/β-catenin inhibitor controlling interneuromast progenitor proliferation should be expressed in wild type Schwann cells until at least 48 hpf, when intercalary neuromasts begin to form. In embryos in which ErbB signaling is abrogated and interneuromast cells proliferate, the Wnt/β-catenin inhibitor should be downregulated. However, the known Wnt/β-catenin inhibitors *sfrp1a*, *wif1*, *dkk1b* and *dkk2* are upregulated in *nrg1-3^{z26}* mutants or *Tg(sox10:DNerbb4)* larvae and *wif1*, *dkk1b* and *dkk2* are not expressed in 48 hpf wild type interneuromast or Schwann cells (*Figure 13*). Therefore, the expression patterns of these inhibitors do not correlate with a function in inhibiting Wnt/β-catenin signaling in 48 hpf wild type interneuromast cells or the release of inhibition in *nrg1-3^{z26}* mutant larvae. Our results are consistent with several recent reports that have shown the importance for Wnt/β-catenin signaling in regulating proliferation in mammalian inner ear support cells and zebrafish neuromasts (*Chai et al., 2012*; *Jacques et al., 2012*; *Jan et al., 2013*; *Shi et al., 2012*; *Head et al., 2013*; *Wada et al., 2013b*). We identified *wnt10a* as the potential ligand required for intercalary neuromast formation, which should be either inhibited by Schwann cells or induced in their absence. *wnt10a* is expressed in interneuromast cells of Schwann cell deficient mutants (*Figure 4A–B*, *Figure 4—figure supplement 1*) and is induced 6 hr post AG1478 treatment (data not shown). However, even though *lef1* and the Wnt/β-catenin reporter are expressed in wild type interneuromast cells at 48 hpf (*Figure 6*), we did not detect *wnt10a* expression, suggesting that additional Wnt ligands upregulate Wnt/β-catenin signaling at this time.

## ErbB receptor tyrosine kinases regulate multiple stem cell niches

Formation of the adult zebrafish pigment pattern depends on ErbB3b, in part non-cell-autonomously, by regulating the formation of dorsal root ganglion neurons, which act as a niche for adult melanophore precursors (*Budi et al., 2011*; *Dooley et al., 2013*). We found that *nrg1-3^{z26}* mutants mimic the *erbb3b* pigment pattern phenotype (*Figure 1—figure supplement 2*). Whether Nrg1-3 is required

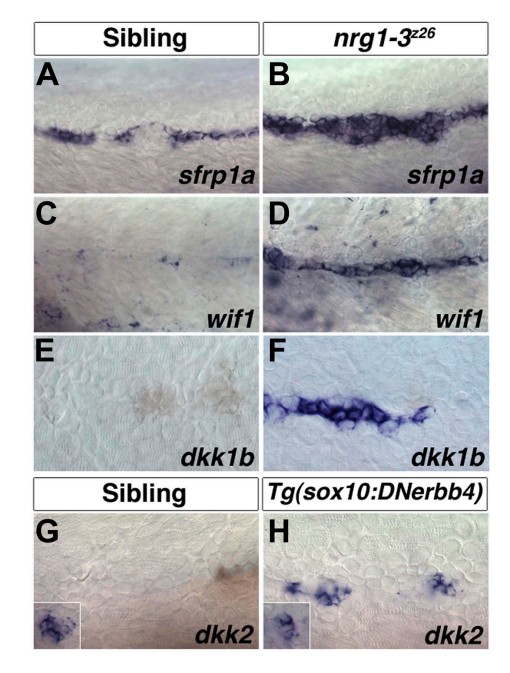

**Figure 13**. Expression of several Wnt/β-catenin inhibitors are increased in *nrg1-3^z26*. Control sibling and *nrg1-3^z26* mutants were fixed at 48 hpf and processed for in situ hybridization. Control larvae show *sfrp1a* expression in interneuromast and mantle cells (**A**). Expression of *sfrp1a* is increased in *nrg1-3^z26* (**B**). Control larvae show no expression of *wif1* (**C**) or *dkk1b* (**F**) in interneuromast or Schwann cells. Both *wif1* (**D**) and *dkk1b* (**E**) are induced in interneuromast cells of *nrg1-3^z26*. (**G**) Control larvae show expression of *dkk2* in primary neuromasts (inset) but not in interneuromast or Schwann cells. (**H**) In *Tg(sox10:DNerbb4) dkk2* is expressed in neuromasts (inset), and is upregulated in interneuromast cells.

for melanophore progenitor niche formation or maintenance is currently unknown. The signaling pathways regulating the niche downstream of ErbB3b have not been identified, but it is interesting to speculate that Wnt/β-catenin signaling may also be involved.

Another ErbB family member, ErbB1, inhibits neural stem cell proliferation non-cell-autonomously through inhibition of Notch activity (*Aguirre et al., 2010*). Notch signaling serves several functions in the development of the lateral line and therefore, is another likely candidate to regulate intercalary neuromast formation. Notch signaling is not active in wild type interneuromast cells but is activated in *nrg1-3^z26* mutants (*Figure 4*). However, the time course expression analysis after ErbB inhibition uncovered that Notch signaling is not upregulated until 24 hr post treatment, well after the initiation of interneuromast cell proliferation (*Figures 3 and 6*, *Figure 4—figure supplement 1*). In addition, pharmacological inhibition of Notch activity, using the γ-secretase inhibitors DAPT or LY411575, did not induce Wnt or Fgf ligand expression (*Figure 4—figure supplement 2*) or intercalary neuromasts in wild type larvae (data not shown). Therefore we conclude that, similar to Fgf, Notch signaling is only playing a later role in intercalary neuromast development, likely regulating the choice between support or hair cell specification as in primary neuromasts (*Itoh and Chitnis, 2001*).

Recently, it was shown that non-myelinating Schwann cells constitute part of a niche, non-cell-autonomously controlling the quiescence of hematopoietic stem cells (HSC) (*Yamazaki et al., 2011*). Removing Schwann cells in mice by axotomy lead to loss of active TGF-β and an increase in HSC proliferation (*Yamazaki et al., 2011*). The TGF-β pathway has been implicated to interact with ErbB signaling (*Chow et al., 2011*). *tgfb1a* is also robustly expressed in the migrating lateral line primordium and we therefore examined if this pathway acts as an intermediary between ErbB and Wnt/β-catenin signaling (ZFIN, http://zfin.org). We treated wild type larvae with two pharmacological inhibitors of Tgf-β receptors, SB505124 or SB431542 (*Hagos and Dougan, 2007*). Neither drug induced an increase in interneuromast number or clusters of interneuromast cells, suggesting that Tgf-β signaling does not inhibit interneuromast cell proliferation and is not involved in their regulation (data not shown).

In *Drosophila*, glial cells control quiescence and proliferation of neural progenitors depending on the developmental stage (*Doe, 2008*). This non-cell-autonomous interaction is regulated by ErbB (*spitz/TGF-α*), HSPGs (*trol/perlecan*), and a secreted glycoprotein, *anachronism* (*Ebens et al., 1993*; *Voigt et al., 2002*; *Morante et al., 2013*). Based on our recent results it would be interesting to examine if these molecules or pathways interact with the Wnt/β-catenin pathway. For example, Perlecan interacts with Wnt/β-catenin and Fgf signaling, however the other pathways have not been examined yet (*Park et al., 2003*; *Kamimura et al., 2013*).

## Developmental and postembryonic intercalary neuromast formation relies on similar but not identical mechanisms

We find that a set of the same signaling pathways are involved in neuromast differentiation from interneuromast cells as in neuromast formation in the primordium. However, primary neuromasts form

within the migrating primordium and are deposited, whereas interneuromast cells proliferate to form a cluster of progenitor cells that eventually differentiate (*Video 1*). Therefore some differences exist. Both Fgf and Wnt/β-catenin are required for proliferation in the primordium (*Aman et al., 2011*), whereas interneuromast proliferation depends on Wnt/β-catenin signaling only. Nevertheless, both neuromast types require Fgf signaling for rosette formation and hair cell specification (*Figure 10A–F; 11E–H* and *Lecaudey et al., 2008*; *Nechiporuk and Raible, 2008*). Fgf signaling is induced by Wnt/β-catenin signaling in the primordium, and likely also in interneuromast cells (*Figure 4—figure supplement 2 and 3*). However, it is not upregulated for many hours after the Wnt/β-catenin pathway is activated post ErbB inhibition (*Figures 5 and 6*) or after BIO treatment (*Figure 4—figure supplement 3*). Also, activation of the Wnt/β-catenin pathway by GSK-3 inhibition does not induce *pea3* in interneuromast cells during the first 24 hr post treatment (data not shown). Therefore, in contrast to the primordium, Fgf may not be a direct target of Wnt/β-catenin signaling in interneuromast cells. Neuromast formation in the primordium occurs in an existing and migrating tissue whereas intercalary neuromasts form from a string of cells. Therefore, it is not surprising that some of the signaling interactions between these two modes of neuromast formations are different.

## ErbB signaling can exert oncogenic or tumor suppressive functions

A detailed analysis of ErbB receptor functions is important as upregulation of ErbB family receptor tyrosine kinases is characteristic of many human cancers, notably breast cancer, resulting in enhanced tumorigenesis. Consequently, ErbB receptors are important therapeutic targets (*Moasser, 2007*). Although cancers typically originate from unique cell types, tumors can be heterogeneous, containing multiple cell types. As the repertoire of pharmacological kinase inhibitors that are being developed to treat cancer increases, it becomes crucial to take into consideration that different ErbB receptor/ligand combinations can result in counterproductive cellular responses. Our results from the zebrafish lateral line stress the importance that ErbB signaling also has non-cell-autonomous, anti-proliferative functions, and that these anti-proliferative functions play a role during the regulation of neuronal progenitor regulation during development.

In conclusion, we identified that ErbB signaling in Schwann cells non-cell-autonomously regulates a molecular signaling network consisting of Wnt/β-catenin, Fgf and Notch pathways within neural progenitors, regulating their quiescence and activation. Given how conserved signaling interactions are during development and across species, it is tempting to speculate that these four pathways also interact in other neural progenitor cell populations.

## Materials and methods

### Fish maintenance and fish strains

We used the following fish strains; *rowgain/erbb2* and *hypersensitive/erbb3b* (*Grant et al., 2005*), *nrg1-3^z26* (*Perlin et al., 2011*), *erbb2^st61* (*Lyons et al., 2005*), *Tg(foxd3:gfp)^zf15* (*Gilmour et al., 2002*), *Tg(clndB:lyngfp)^zf106* (*Haas and Gilmour, 2006*), *Et(krt4:EGFP)^sqet20* (*Parinov et al., 2004*), *Tg(hsp70l:dkk1b-GFP)^w32* (*Stoick-Cooper et al., 2007*), *Tg(hsp70l:dnfgfr1-EGFP)^pd1* (*Lee et al., 2005*), *Tg(OTM:d2EGFP)^kyu1* referred to as *Tg(Tcf/Lef-miniP:dGFP)* (*Shimizu et al., 2012*), *cntnap2a^nkhgn39dET* referred to as HGN39D (*Nagayoshi et al., 2008*) and *lef1^zd11* (*Wang et al., 2012*). To generate the *Tg(sox10:DNhsaerbb4-RFP)*, we used the zebrafish Tol2 kit (*Kwan et al., 2007*). We cloned the zebrafish 7.2 kb *sox10* promoter, obtained from Thomas Carney, into the 5′ entry vector. Human dominant-negative ErbB4 (hsaDNerbb4) construct was a gift of Gabriel Corfas (*Rio et al., 1997*). The flag tag on hsaDNerbb4 was replaced with mRFP and cloned into the middle entry vector. To generate the final vector the 5′ entry, middle entry and 3′ polyA entry vector were recombined with the destination vector containing the *cmcl2:gfp* expression cassette, in order to identify transgenics based on GFP expression in the heart (*Kwan et al., 2007*). The recombined vector was then injected into one cell stage embryos from the Tubingen strain along with transposase mRNA. Founder GFP positive heart carriers were raised and identified by crossing to wild type fish. The positive carriers have been maintained over three generations.

### In situ hybridization

In situ hybridization was performed as previously described (*Kopinke et al., 2006*). The following probes were used *lef1, fgf3, fgf10, fgfr1a, pea3, atoh1a, dkk1b, klf4* (*Aman and Piotrowski, 2008*), *sfrp1a* (*Tendeng and Houart, 2006*), *mbp* (*Brosamle and Halpern, 2002*), *dkk2* (*Wada et al., 2013b*).

To clone additional probes the following primers were used, for *myca;* forward 5'-ggtcctggacactccaccta-3' and reverse 5'-atgcactctgtcgccttctt-3', *beta-catenin-2 (ctnnb2);* forward 5'-cgactctgctcatccaacaa-3' and reverse 5'-aggatctgcaggcagtctgt-3', *wnt10a;* forward 5'-cttcagcaggggtttcagag-3' and reverse 5'-tccctggctggtcttgttac-3', *wif1;* forward 5'-aaccaaaggatggxtttcagg-3' and reverse 5'- aggtttaaaccacatagtt-ggtttcag-3'. For low magnification images, individual images were stitch together using ImageJ.

## Alkaline phosphatase and DASPEI staining

For alkaline phosphatase staining larvae were fixed overnight at room temperature in 4% paraformaldehyde. Larvae were then washed three times 5 min each in PBS/0.3% Tween-20 followed by three 5-min washes in staining buffer (50 mM $MgCl_2$, 100 mM NaCl, 100 mM Tris pH 9.5 and 0.1% Tween-20). Larvae were then placed in staining buffer plus NBT/BCIP (Roche, USA) and stained at room temperature in the dark. The staining reaction was stopped with 4% paraformaldehyde. To label hair cells embryos were placed in a 0.06 mg/ml solution of DASPEI (2-(4-(dimethylamino)styryl)-N-ethylpyridinium iodide, [Invitrogen, USA]) diluted in embryo media for 10 min. Embryos were then briefly washed and anesthetized with Tricane for imaging under a dissecting fluorescent microscope. For low magnification images, individual images were stitch together using ImageJ.

## Transplantation assay

Transplantation assays were performed as previously described (*Aman and Piotrowski, 2008*). Donor embryos were injected with 5% Alexa-568 and 3% lysine-fixable biotinylated-dextran (Invitrogen, USA) at the one cell stage. *Tg(foxd3:gfp)* or *Tg(clndB:lyngfp)* wild type donors were used to test rescue of *erbb2* mutant hosts. *Tg(clndB:lyngfp)* donors were used to test rescue of *nrg1-3$^{z226}$* mutant hosts. Host embryos were screened for lateral line or Schwann cell clones at 24 hr post fertilization (hpf). Embryos were imaged at 48 hpf to record the extent of the transplanted clone. To identify mutant hosts or donors, neuromasts were counted after DASPEI staining at 4 days post fertilization (dpf). Both transplanted and untransplanted sides of the hosts were imaged with a Zeiss LSM 510 or 780 confocal microscope at 20X. For low magnification images, individual images were stitch together using ImageJ.

## Pharmacological inhibitors

All chemical inhibitors were added to embryo media with a final concentration of 1% DMSO. Negative controls consisted of 1% DMSO only. The ErbB inhibitor AG1478 was used at 3 µM. The Fgf receptor inhibitors PD173074 or SU5402 were added at 100 µM or 10 µM respectively. The Wnt/β-catenin inhibitors IWR-1 or XAV939 were used at 40 µM and 20 µM respectively. The GSK-3 inhibitor BIO was used at 2 µM. The γ-secretase inhibitor LY411575 was used at 100 µM. LY411575 was obtained from Santa Cruz (USA) and all other inhibitors were purchased from Tocris (USA).

## BrdU assay

BrdU incorporation was performed in *Tg(foxd3:gfp)* or *Tg(SqET20:gfp)* transgenics by addition of 10 mM BrdU (Sigma, USA) with 1% DMSO or chemical inhibitors for various lengths of times as indicated in the text. Embryos were fixed in 4% paraformaldehyde overnight. BrdU immunostaining was performed as described except embryos were treated for 15 min with proteinase K (*Aman et al., 2011*). To visualize GFP, larvae were also immunostained with rabbit anti-GFP (Invitrogen, USA) at 1/400 dilution. All embryos were counterstained with DAPI (Invitrogen, USA). To quantify BrdU index we counted BrdU and GFP double positive cells between the first and second deposited neuromasts. Immunostained embryos were imaged with a Zeiss LSM 510 or 780 confocal microscopes at 40X.

## Heat shock induction of gene expression

Heat shock induction was done at various developmental ages as indicated in the text. Embryos were placed at 39°C for 20 min, room temperature for 20 min and then another 20 min at 39°C. Embryos were then allowed to develop at 28.5°C.

## Time-lapse imaging

Time-lapse imaging was performed similar as described (*Aman and Piotrowski, 2008*). *Tg(SqET20:gfp)* or *Tg(SqET20:gfp)/Tg(sox10:DNhsaerbb4-RFP)* were anesthetized with Tricaine and mounted in 0.8% low melting agarose on glass bottom dishes (MatTek, USA). Embryos were imaged with a Zeiss 710 or 780 confocal microscopes using a 40X water objective in a climate-controlled chamber set to 28°C.

## Acknowledgements

We are grateful to Dr Gabriel Corfas for the human dominant negative ErbB4 plasmid, Dr Thomas Carney for the Sox10 promoter and Drs Chi-Bin Chen and Kristen Kwan for the Tol2-kit. We thank the zebrafish cores of both the University of Utah and the Stowers Institute for Medical Research for excellent zebrafish husbandry. We thank Drs Rich Dorsky, William Talbot, Darren Gilmour, Vladimir Korzh, Randall Moon, Kenneth Poss and Tohru Ishitani for providing zebrafish strains and Dr Robb Krumlauf and Marina Venero-Galanternik for valuable comments on the manuscript and Hua Li for help with statistics. We thank Dr Andy Aman and Robert Duncan for cloning of in situ probes, MinhTu Nguyen and Megan Smith for technical help, Joshua Sasine for help with pilot experiments and members of the Piotrowski laboratory for helpful discussions. We would also like to thank the reviewers for valuable suggestions and Mark Miller for help with graphic design.

## Additional information

### Funding

| Funder | Grant reference number | Author |
| --- | --- | --- |
| Huntsman Cancer Institute Multidisciplinary Cancer Research Training Program Grant, University of Utah | 5T32CA093247 | Mark E Lush |
| Institutional Support from the Stowers Institute for Medical Research | | Tatjana Piotrowski |

The funders had no role in study design, data collection and interpretation, or the decision to submit the work for publication.

### Author contributions

MEL, Conception and design, Acquisition of data, Analysis and interpretation of data, Drafting or revising the article; TP, Conception and design, Analysis and interpretation of data, Drafting or revising the article

### Ethics

Animal experimentation: This study was performed in strict accordance with the recommendations in the Guide for the Care and Use of Laboratory Animals of the National Institutes of Health. All animals were handled according to approved institutional animal care and use committee protocols of the University of Utah and the Stowers Institute for Medical Research (IACUC animal protocol number 2011-0080).

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
