## [Decision Letter]

Thank you for sending your work entitled “ErbB expressing Schwann cells control lateral line progenitor cells via non-cell-autonomous regulation of Wnt/β-catenin” for consideration at *eLife*. Your article has been favorably evaluated by a Senior editor and 2 reviewers, one of whom is a member of our Board of Reviewing Editors.

The Reviewing editor and the other reviewer discussed their comments before we reached this decision, and the Reviewing editor has assembled the following comments to help you prepare a revised submission.

This work adds to our understanding of the mechanism of formation of intercalary neuromasts in the zebrafish lateral line system. The authors show that Wnt signaling-mediated proliferation in interneuromast cells is an early step in intercalary neuromast formation from these cells, which serve as a quiescent progenitor population. The study is a nice follow up of previous work from this group and others, which demonstrated a role for ErbB signaling in preventing premature intercalary neuromast formation. The figures are clear and easy to follow, and in general, the study is careful and thorough.

The following points should be addressed before publication:

1) Wnt signalling in the interneuromast cells. It is very interesting that the authors describe the expression of *wnt10a* in the context of intercalary neuromast formation, especially since no Wnt homologue has yet been shown to play a clear role primary neuromast formation. However, the role of *wnt10a* has not been tested rigorously. Which cells express it? Which signaling pathway determines its expression? Though obviously its expression contributes to Wnt signaling, is its transcription dependent on Wnt, FGF or Notch signaling?

2) Is the pattern of induced expression of *fgf3* and *fgf10* consistent with its dependence on Wnt signaling as in the leading end of the primordium or is it determined by *atoh1a* expression?

3) Which cells express *her4.1*?

4) Why does Wnt signaling reduce prematurely following AG1478? Does the down regulation of Wnt activity correlate with an upregulation of some other signaling system at this time?

5) Continued role of ErbB signalling in Schwann cells. The data supporting this claim (Figure 1—figure supplement 4) are not entirely clear. A graph of cell counts indicates a significant difference between 0.035 and 0.025 cells/µm (Figure 1—figure supplement 4), and the methods indicate that counts of cells were taken between the first and second neuromasts. In the first place, the distance between first and second neuromasts will be very different between wild-type and inhibitor-treated embryos (and can be variable even in wild-type). The authors should state the actual distance and the numbers of cells counted, which must be quite small. Later experiments merely seem to confirm the requirement for Schwann cells, rather than the requirement for continued ErbB signalling in Schwann cells, and so this conclusion should be toned down. In particular, the authors should remove the words 'ErbB signaling in' from the sentence “In conclusion, the absence of ErbB signaling in Schwann cells, leads first to interneuromast cell proliferation…”, as this particular experiment uses a transgenic line in which Schwann cells are depleted.

6) Neuregulin mutant. The authors should change the wording of the sentence:

“Here we show that a mutation in *neuregulin 1-3 (nrg1-3)* also exhibits precocious neuromast formation”. The authors have previously reported that this mutant was isolated through a screen for supernumerary neuromasts, so this finding was effectively reported in their previous publication (Perlin et al., 2011), which is not cited on this page.

7) Statistical tests. Student's t-tests are the only statistical tests mentioned in the manuscript, but there are a number of instances throughout (e.g., control vs multiple experimental treatments) where t-tests would not be appropriate. All the statistical tests should be re-evaluated carefully to make sure that appropriate tests are chosen for each experimental situation.

8) A reference for the PD173074 Fgf inhibitor should be given. This is not widely used, and it is surprising that it affects the lateral line ganglion so differently to SU5402 (Figure 10—figure supplement 1). The authors state that “likely, the two inhibitors affect a different set of Fgf receptors or pathways”, but this is not backed up with a citation or other precedent.

---

## [Author Response]

*1) Wnt signalling in the interneuromast cells. It is very interesting that the authors describe the expression of* wnt10a *in the context of intercalary neuromast formation, especially since no Wnt homologue has yet been shown to play a clear role primary neuromast formation. However, the role of* wnt10a *has not been tested rigorously. Which cells express it? Which signaling pathway determines its expression? Though obviously its expression contributes to Wnt signaling, is its transcription dependent on Wnt, FGF or Notch signaling*?

*Which cells express* wnt10a?

We have now included data showing that *wnt10a* is expressed in proliferating interneuromast cells only (Figure 4—figure supplement 1, described within the paragraph ‘Wnt/β-catenin, Fgf and Notch signaling pathways are upregulated in ErbB/Neuregulin pathway mutants’). To determine which cells express *wnt10a* we imaged *Tg(ET20:gfp)* siblings and *Tg(ET20:gfp)*/*Tg(sox10:DNerbb4)* larvae at 48hpf, then fixed and performed in situ hybridization for *wnt10a* on the same animals*.* We only show the data for *Tg(ET20:gfp)*/*Tg(sox10:DNerbb4)* larvae because, as shown in Figure 4, *wnt10a* is not expressed in control interneuromast cells. *wnt10a* is expressed within interneuromast cells of *Tg(sox10:DNerbb4)* but is downregulated in differentiating neuromasts.

*Which signaling pathways activate* wnt10a?

We have now determined that *wnt10a* transcription is dependent on Wnt/β-catenin activity but is not dependent on Fgf or Notch signaling (Figure 6–figure supplement 1A-H, described within the paragraph ‘Wnt and Fgf ligands are dependent on Wnt/β-catenin signaling’). To test which signaling pathways regulate *wnt10a* expression we treated control sibling or *Tg(sox10:DNerbb4)* embryos at 32hpf (hours post fertilization) with Wnt/ β-catenin, Fgf or Notch signaling inhibitors, and subsequently fixed them at 48hpf and performed in situ hybridization. *wnt10a* is downregulated in *Tg(sox10:DNerbb4)* embryos by inhibition of Wnt/β-catenin signaling but not by inhibition of Fgf or Notch signaling (Figure 6–figure supplement 1, E-H).

We also tested if activation of the Wnt/β-catenin pathway with BIO leads to upregulation of *wnt10a* transcription in interneuromast cells. Wild type embryos were treated with DMSO or BIO at 48hpf then fixed at 6, 12 or 24 hours post-treatment. The known Wnt/β-catenin target gene *lef1* was induced at all time points (Figure 6–figure supplement 2, A-F, described within the paragraph ‘Wnt and Fgf ligands are dependent on Wnt/β-catenin signaling’). Likewise *wnt10a* was induced from 6-24 hours post treatment (Figure 6–figure supplement 2, G-L, described within the paragraph ‘Wnt and Fgf ligands are dependent on Wnt/β-catenin signaling’). These results support the conclusion of the loss of function studies that Wnt/β-catenin signaling acts upstream of *wnt10a*, whereas Fgf and Notch signaling do not affect *wnt10a* expression.

*2) Is the pattern of induced expression of* fgf3 *and* fgf10 *consistent with its dependence on Wnt signaling as in the leading end of the primordium or is it determined by* atoh1a *expression?*

We have now performed experiments that reveal that in interneuromast cells Fgf signaling depends on Wnt/β-catenin signaling as in the leading region of the primordium. However, Fgf ligand activation takes 12-24h suggesting that Fgf signaling might not be a direct target. To determine which signaling pathways induce *fgf3* and *fgf10*, we treated 32hpf control siblings or *Tg(sox10:DNerbb4)* embryos with Wnt/β-catenin, Fgf or Notch signaling inhibitors. We subsequently fixed the embryos at 48hpf and performed in situ hybridization with *fgf3* and *fgf10*. Both *fgf3* and *fgf10* expression are reduced in *Tg(sox10:DNerbb4)* embryos after blocking Wnt/β-catenin signaling but not after inhibition of Fgf or Notch signaling (Figure 6–figure supplement 1, M-P and U-X, described within the paragraph ‘Wnt and Fgf ligands are dependent on Wnt/β-catenin signaling’).

Our new experiments also demonstrate that Wnt/β-catenin activation induces transcription of *fgf3* and *fgf10* in interneuromast cells (Figure 6–figure supplement 2M-X, described within the paragraph ‘Wnt and Fgf ligands are dependent on Wnt/β-catenin signaling’). We treated wild type siblings with DMSO or BIO at 48hpf and fixed the larvae at 6, 12 or 24 hours post-treatment. *fgf10* is induced by 12 hours of BIO treatment while *fgf3* is only induced at 24 hours post treatment. This relatively long time span between Wnt/β-catenin activation and induction of *fgf* ligands suggests that Fgf signaling might not be a direct target of Wnt/β-catenin signaling in interneuromast cells.

*3) Which cells express her4.1*?

We imaged 48pf *Tg(ET20:gfp)* siblings and *Tg(ET20:gfp)*/*Tg(sox10:DNerbb4)* larvae and fixed and performed in situ for *her4.1.* We only show the data for *Tg(ET20:gfp)*/*Tg(sox10:DNerbb4)* larvae because, as shown in Figure 4, *her4.1* is not expressed in control interneuromast cells. In *Tg(ET20:gfp)*/*Tg(sox10:DNerbb4)* larvae *her4.1* is expressed in deposited neuromasts (Figure 4—figure supplement 1, described within the paragraph ‘Wnt/β-catenin, Fgf and Notch signaling pathways are upregulated in ErbB/Neuregulin pathway mutants’) and also in some cells within newly differentiating neuromasts (Figure 4—figure supplement 1, yellow squares). However, *her4.1* is not expressed in proliferating interneuromast cells (Figure 4—figure supplement 1, yellow arrowheads).

*4) Why does Wnt signaling reduce prematurely following AG1478? Does the down regulation of Wnt activity correlate with an upregulation of some other signaling system at this time*?

As noted by the reviewer Wnt/β-catenin signaling is upregulated at 6 and 12 hours post treatment but returns close to wild type levels at 24 hours post AG1478 treatment (Figure 6). It is possible that other signaling pathways are induced at this later stage and are responsible for inducing a Wnt/β-catenin inhibitor in interneuromast cells. One Wnt/β-catenin inhibitor, *dkk1b,* is an Fgf target in the primordium (Aman and Piotrowski, 2008). However, Fgf signaling, as judged by *pea3* expression, is not induced until after 24 hours of AG1478 treatment. We also performed an expression analysis with *dkk1b* and *dkk2* after AG1478 addition and neither gene was induced by 36 hours post treatment (data not shown), illustrating that *dkk1b* and *dkk2* are likely not the responsible Wnt/β-catenin signaling inhibitors at the AG1478 24 hour time point. This suggest that if Fgf is also regulating *dkk* expression during intercalary neuromast formation, as it is in the primordium, it is acting at a time point later than 24 hours post AG1478 treatment. Therefore, Fgf signaling is not a likely candidate to cause the decrease in Wnt/β-catenin signaling (Figure 6).

Based on our experiments with the Notch inhibitor LY411575 (Figure 6–figure supplement 1D,L,T), and the absence of Notch signaling in wild type interneuromast cells, we concluded that Notch signaling is not required for keeping interneuromast cells quiescent by inhibiting Wnt/β-catenin signaling in wild type embryos. However, Notch signaling, as judged by *her4.1* expression, is induced at 24 hours after AG1478 and could therefore be responsible for the upregulation of a Wnt/β-catenin inhibitor at this stage (Figure 6). To investigate if Wnt/β-catenin signaling persists if Notch signaling is inhibited at 24 hour post AG1478 treatment we soaked embryos simultaneously in AG1478 and LY411575. Unfortunately, the combination of these two drugs had toxic effects on the embryos. We think this down regulation of Wnt/β-catenin signaling is interesting; however, it would require more elaborate experiments that are beyond the focus of this study.

*5) Continued role of ErbB signalling in Schwann cells. The data supporting this claim (*Figure 1—figure supplement 4*) are not entirely clear. A graph of cell counts indicates a significant difference between 0.035 and 0.025 cells/µm (*Figure 1—figure supplement 4*), and the methods indicate that counts of cells were taken between the first and second neuromasts. In the first place, the distance between first and second neuromasts will be very different between wild-type and inhibitor-treated embryos (and can be variable even in wild-type). The authors should state the actual distance and the numbers of cells counted, which must be quite small. Later experiments merely seem to confirm the requirement for Schwann cells, rather than the requirement for continued ErbB signalling in Schwann cells, and so this conclusion should be toned down. In particular, the authors should remove the words 'ErbB signaling in' from the sentence “In conclusion, the absence of ErbB signaling in Schwann cells, leads first to interneuromast cell proliferation…”, as this particular experiment uses a transgenic line in which Schwann cells are depleted*.

We counted Schwann cells between the first and second deposited neuromasts, L1 and L2. We re-worded the methods section to make this clearer. We agree that the distance between neuromasts L1 and L2 varies and therefore we chose to calculate the number of Schwann cells per μm to control for the variation in distance between neuromasts. We have added within the figure legend the average number of Schwann cells counted for each group (Figure 1—figure supplement 4). At 6 hours post DMSO treatment we counted an average of 12 Schwann cells and the average distance between L1 and L2 was 531μm. In AG1478 treated larvae we counted on average 13 Schwann cells over 546 μm. At 14 hours post DMSO treatment we counted an average of 20 Schwann cells over 576 μm and 14 Schwann cells over 536 μm after AG1478 treatment. At 24 hours post DMSO treatment we counted an average of 14 Schwann cells over 511 μm and 6 Schwann cells over 508 μm after AG1478 treatment.

We have removed 'ErbB signaling in' from the sentence indicated, as we agree the experiment described is examining Schwann cell deficient zebrafish.

*6) Neuregulin mutant. The authors should change the wording of the sentence*:

*“Here we show that a mutation in neuregulin 1-3 (nrg1-3) also exhibits precocious neuromast formation”. The authors have previously reported that this mutant was isolated through a screen for supernumerary neuromasts, so this finding was effectively reported in their previous publication (Perlin et al., 2011), which is not cited on this page*.

We have deleted this sentence from the manuscript and added *nrg1-3* to the preceding sentence of the manuscript.

*7) Statistical tests. Student's t-tests are the only statistical tests mentioned in the manuscript, but there are a number of instances throughout (e.g., control vs multiple experimental treatments) where t-tests would not be appropriate. All the statistical tests should be re-evaluated carefully to make sure that appropriate tests are chosen for each experimental situation*.

We have re-evaluated the statistical analysis and have performed a one-way ANOVA with Tukey pairwise comparison for experiments that have multiple experimental treatments. This new analysis was performed on data graphed in Figures 7 and 10, Figure 1—figure supplement 1 and Figure 10—figure supplement 1. For two of these experiments the results are now no longer significant, as the p-value is greater than 0.05. One of these experiments is the test of neuromast formation after AG1478 is added at different developmental time points (Figure 1—figure supplement 1). The 72 hour time point is no longer significantly different from the DMSO treated group. We have updated the text and figure to reflect this change. This does not change the main conclusion of this experiment which is AG1478 induces an increase in neuromast number, even if given after Schwann cell migration is completed at 50 or 59 hpf.

The second experiment is described in Figure 10. The difference in BrdU incorporation within *Tg(SqET20:gfp)* cells between DMSO and SU5402 treated control siblings is not significant. We have updated the text to reflect this change. Again this does not change the conclusion of the experiment that Fgf pathway inhibition, either by SU5402 or PD173074, does not block interneuromast cell proliferation in *Tg(sox10:DNerbb4)* fish.

For the remaining experiments we continue to use a Student’s t-test**,** as we are comparing only two groups**.** We now state within the figure legends which statistical analysis was used for each graph.

*8) A reference for the PD173074 Fgf inhibitor should be given. This is not widely used, and it is surprising that it affects the lateral line ganglion so differently to SU5402 (*Figure 10—figure supplement 1*). The authors state that “likely, the two inhibitors affect a different set of Fgf receptors or pathways”, but this is not backed up with a citation or other precedent*.

We have not found a reference that directly compares the two different FGFR inhibitors with different FGF receptors and we therefore only reference the original paper that describes PD173074 (49). As mentioned in the text, the fact that PD173074 causes an increase in posterior lateral line ganglion cell number matches the results the Riley laboratory acquired using a heat-shock dominant negative FGFR transgenic in the zebrafish statoacoustic ganglion (70). As dominant negative Fgfr is likely a more specific inhibitor than pharmacological inhibitors we suspect that PD173074 is more specific in blocking Fgf signaling within the posterior lateral line ganglion than SU5402 that is possibly affecting other receptor tyrosine kinases. We would like to emphasize that this difference in drug effect was only seen in the posterior lateral line ganglion. SU5402 and PD173074 induced the same phenotypes during primordium migration (data not shown) as well as intercalary neuromast formation described in this paper.